# Physical-controlled $CO_2$ effluxes from reservoir surface in the upper Mekong River Basin: a case study in the Gongguoqiao Reservoir

Lin Lin[1], Xixi Lu[1, 2, *], Shaoda Liu[3], Shie-Yui Liong[4] and Kaidao Fu[5, *]

[1]Department of geography, National University of Singapore, 117570, Singapore
[2]Inner Mongolia Key Lab of River and Lake Ecology, School of Ecology and Environment, Inner Mongolia University, Hohhot, Inner Mongolia, 010021, China
[3]Yale School of Forestry & Environmental Studies 195 Prospect Street New Haven, CT 06511. USA
[4]Tropical Marine Science Institute (TMSI), National University of Singapore, 117570, Singapore
[4]Asian International River Center, Yunnan University, Chenggong University City, Chenggong, Kunming, Yunnan, 650500,
China

*Correspondence to*: Xixi Lu (geoluxx@nus.edu.sg), Kaidao Fu (kdfu@ynu.edu.cn)

**Abstract.** Impounding alters the carbon transport in rivers. To quantify this effect, we measured $CO_2$ effluxes from a mountainous valley-type reservoir in the upper Mekong River (known as Lancang River in China). $CO_2$ evasion rates from
the reservoir surface were $408 \pm 337$ mg $CO_2$ m$^{-2}$ d$^{-1}$ in the dry season and $305 \pm 262$ mg $CO_2$ m$^{-2}$ d$^{-1}$ in the rainy season, respectively, much lower than those from the riverine channels ($1567 \pm 2312$ mg $CO_2$ m$^{-2}$ d$^{-1}$ at the mainstem and $905 \pm 1536$ mg $CO_2$ m$^{-2}$ d$^{-1}$ at the tributary). Low effluxes in pelagic area were caused by low allochthonous organic carbon (OC) inputs and photosynthetic uptake of $CO_2$. Negative relationship between $CO_2$ efflux and water temperature suggested $CO_2$ emissions at the pelagic area were partly offset by photosynthesis in the warmer rainy season. $CO_2$ emissions from the
reservoir outlet and littoral area, which were usually considered as hotspots of $CO_2$ emissions, had a low contribution to the total emission because of epilimnion water spilling and small area of the littoral zones. Yet at the river inlets effluxes were much higher in the dry season than in the rainy season because different mixing modes occurred in the two seasons. When the river joined the receiving waterbody in the dry season, warmer and lighter inflow became an overflow and large amounts of $CO_2$ were released to the atmosphere as the overflow contacted the atmosphere directly. Extended water retention time
due to water storage might also help mineralization of OC. In the wet season, however, colder, turbid and heavier inflow plunged into the reservoir and was discharged to the downstream for hydroelectricity, leaving insufficient time for decomposition of OC. Besides, diurnal efflux variability indicated that the effluxes were significantly higher in the nighttime than in the daytime, which increased the estimated annual emission rate by half.

# 1 Introduction

Supersaturation of $CO_2$ in the inland waters (Cole et al., 1994) resulted in substantial carbon outgassing to the atmosphere (Battin et al., 2009; Cole et al., 2007; Raymond et al., 2013; Tranvik et al., 2009). Loss of carbon to the atmosphere from inland waters has been recognized as an important part of carbon cycling which faced great anthropogenic impacts (Maavara et al., 2017; Regnier et al., 2013). Damming rivers for water supply, irrigation, hydroelectricity and flood controls is one of the most drastic changes in inland waters (Lehner & Döll, 2004; Varis et al., 2012; Yang & Lu, 2014). By flooding large area of forests, soils and different kinds of organic matter, reservoirs have been identified as a large potential carbon source to the atmosphere since last century and have caused a serious perturbation to the global carbon budget (Fearnside, 1997; Kelly et al., 1994; Rudd et al., 1993). Damming rivers not only enlarged the water surface, but also produced more greenhouse gases (GHGs), mainly carbon dioxide and methane, than the natural waterbodies (Barros et al., 2011, Deemer et al., 2016, Mendonça et al., 2012a). Most of the carbon was released in the form of carbon dioxide, even though methane made up a bigger part of the warming potential (Deemer et al., 2016, Demarty and Bastien, 2011).

Efforts have been made to evaluate $CO_2$ emissions from reservoir surfaces (Raymond et al., 2013; Varis et al., 2012; Vincent et al., 2000) and accumulated case studies indicated that $CO_2$ emission rates exhibited great seasonal variability and spatial heterogeneity (Barros et al., 2011; Deemer et al., 2016). Quantity and quality of dissolved organic carbon (DOC) and water temperature were considered as the most important factors that controlled the $CO_2$ fluxes from reservoirs (Mendonça et al., 2012a; Tadonleke et al., 2012). Young tropical reservoirs and the reservoirs with substantial labile OC tended to have higher emission rates (Barros et al., 2011; Fearnside, 1997). However, in China, the country with the most reservoirs (89,696 reservoirs) in the world (Yang & Lu, 2014; Yang et al., 2016), analysis on $pCO_2$ indicated that most of the effluxes from reservoir surface were much lower than that from tropical and boreal reservoirs (Li & Zhang, 2014; Li et al, 2015; Liu et al., 2016b; Ran et al., 2017). Lower effluxes in the reservoir center (Gao et al., 2017; Mei et al., 2011; Liu et al., 2016b; Liu et al., 2017) implied that the $CO_2$ in reservoir surface was subject to photosynthetic uptake of phytoplankton (Ran et al., 2017; Ran et al., 2018). $CO_2$ Effluxes and $pCO_2$ in reservoirs were regulated by the balance between respiration and photosynthesis, and sensitive to the monsoon climate due to the seasonal variation of water temperature and hydrological condition (Guo et al., 2011; Mei et al., 2011). For example, in the Three Gorges Reservoir, the largest reservoir in China, $CO_2$ emissions from the littoral zone were subjected to the seasonal flooding (Chen et al., 2009; Yang et al., 2012), while the carbon uptake of algae in the stagnant tributaries as a result of heavy eutrophication was heavily influenced by the seasonal variation of hydrological condition (Jiang et al., 2012, Guo et al., 2011, Ran et al., 2011, Zhao et al., 2013).

Despite the spatial heterogeneity (Li & Zhang, 2014), the previous research mostly focused on the reservoirs in the highly populated eastern plain where waterbodies were suffering from heavy eutrophication (Li & Zhang, 2014; Mei et al., 2011). In the less populated Southwestern China where two-thirds of the exploitable hydropower were found and many more

reservoirs are being built, however, the dynamics underlying $CO_2$ emissions has been less understood (Hu & Cheng, 2013). Rivers originated from the Tibetan Plateau flow through the mountainous area of Southwestern China and receive flows from melted glaciers and rainfalls brought by the South Asian monsoon. The precipitation in summer and autumn accounted for 50% and 27% of the annual rainfalls, respectively, producing high waterflow in the warm rainy season. It was supposed

that the $CO_2$ emissions of these rivers were more sensitive to the monsoon climate which regulated rainfalls, nutrient availability, and water discharge. However, the river flows were also regulated by the dams. In particular, dams completed in the upper basin of the Mekong River (or the Lancang River), one of the most important rivers in Southeast Asia, have already affected the hydrological condition, sediment transportation and the $CO_2$ emissions (Lu and Siew, 2006; Lu et al., 2014).

In this study, the Gongguoqiao Reservoir (GGQ), the uppermost reservoir in the Lancang cascading reservoir, was selected as a site for the investigation of the seasonal variation of carbon effluxes. This research aimed at measuring the $CO_2$ evasion with static chamber method and analyzing the spatial heterogeneity, seasonal variation and diurnal variation of the $CO_2$ efflux, in order to examine the mechanism that controlled the $CO_2$ effluxes under the monsoon climate and the damming

effect on carbon emissions. Considering there are seven completed dams in the upper Mekong Basin and another fourteen dams are either under construction or planned, clarifying the coupling effect of the monsoon climate and damming on the $CO_2$ emissions would help us understand the role of reservoirs in the global carbon cycle.

## 2 Methods

### 2.1 Study area

The Gongguoqiao Reservoir (GGQ) is located in Gongguo Town (25º35'9.87"N, 99º20'5.55"E) in Dali Prefecture (Yunnan, China) (Fig. 1). With a catchment area of 97,200 $km^2$, around 32 billion $m^3$ of water flowed into the reservoir annually. The monthly water discharge of inflow to the GGQ Reservoir in 2016 during the study is shown in Fig. 2. Point L (Jiuzhou) was considered the point dividing the upper and middle reach of the Lancang River (Fig. 1). The area was subject to a subtropical monsoon climate; over 80% of the annual rainfalls, 78.6% of the annual water discharge and 95% of the annual sediments

loads to the reservoir occurred in the rainy season spanning from May to October (He and Tang, 2000) (Fig.2). The annual precipitation was 804.90 mm and the monthly average air temperature ranged from 7.6 ℃ to 21.6 ℃, with an annual average of 17.8 ℃ (Fig. S1). There were several villages scattered along the riverside. Before the reservoir filling, the average vegetation cover was only 25% in the steep slope due to intense agricultural activities (Hu, 2010, Xu et al., 2003). The reservoir was filled in September 2011 and had been the uppermost cascading reservoir in the upper Mekong River Basin

until the end of 2016 when the Miaowei Reservoir was filled at its upstream. The outflow from the GGQ feeds the Xiaowan Reservoir at the downstream. The backwater area stretched 44.3 km along the mainstem and 7 km along the tributary, the

Bijiang River, respectively. The width of the reservoir ranged from 110 to 120 m in the dry season. The normal water level was 1307 m, corresponding to a storage of 0.316 billion $m^3$. The reservoir released the top water (in epilimnion zone around 4~5m deep) for hydropower production and generated 4.041 billion kW/h hydroelectricity annually. The reservoir is a daily-operated reservoir due to its small operating capacity (49 million $m^3$). Thus, the water level fluctuated frequently and the average water retention time was 1.4 days. Stratification was found in summer and autumn, but it was generally interrupted by the subsurface flow (unpublished data of this research).

## 2.2 Study methods

### 2.2.1 Sampling

Five sampling points were selected along the mainstem and two from the Bijiang River, a turbid tributary joining the reservoir about 1km before the dam (Fig. 1). The sampling points where the surface velocity could be detected (v>0m/s) were defined as river channels. The average flow velocity was 0.2m/s and 0.7m/s at Point R1 and R2, respectively. Thus the two points were considered as river channels and the flows in channels were regarded as the inflows to the reservoir. Even though the Miaowei Reservoir under construction during the sampling period might have affected the deposition processes of the river, since the water was not impounded and regulated by the dam, Point R1 was considered as pristine river channel. Another point was selected for comparison at the downstream of the dam (Point D) where the flow was regarded as the outflow. The surface flow velocities at all the other points were almost zero and defined as the points in the reservoir. Among the points in the reservoir, Points P1~P4 were defined as pelagic points as they were permanently flooded. Point L was defined as littoral zone with daily flooding and draining owing to the frequent fluctuation of the water level. The point was in a relatively flat wetland formed by the deposited fine sediments.

The sampling campaign started in January 2016. The first two campaigns were carried out in January and March. Samples were collected only in riverine channels, including Point R1, R2 and D. The formal campaigns were conducted twice a month from April to December 2016 before the impounding of the Miaowei Reservoir at the upstream. Samples were collected from 9am to 4pm when sunlight was available and each campaign lasted two to three days. The emission rates were measured following the same order among sampling points. We were not able to collect the samples at the Point L in late October as it was dried out due to a low water level. Totally 127 samples were collected in 16 formal campaigns. For the diurnal variation in fluxes, discontinuous samplings were conducted in the riverine sites during the first sampling campaign in January, while the continuous diel sampling on $CO_2$ effluxes was conducted at a permanently flooded point adjacent to Point L before the last sampling campaign.

The effluxes were measured in situ with a floating chamber connected to a non-dispersive infrared $CO_2$ analyzer (S157-P 0-2000ppm, Qubit, Canada) via the LQ-MINI interface (Vernier, USA). The chamber was a 20cm x 12cm x 10cm

polypropylene rectangle translucent box inserted through a diamond-shape Styrofoam collar. Before measurements it was turned upside down three times to mix the gas within the box. The $CO_2$ analyzer could detect the partial pressure of $CO_2$ ($pCO_2$) down to 1ppm and it was calibrated before the sampling campaigns started. The measurement of $CO_2$ concentration did not begin until the reading of the analyzer became stable at around 400~500ppm. The chamber was fixed to the piles while floating on the water surface.

Calculation of effluxes was based on the slope of graph of concentration versus time according to methodology proposed by Tremblay et al. (2005).

$$Efflux = \frac{slope \times volume}{surface},$$ (1)

where *volume* refers to the air trapped in the chamber and *surface* is the surface of the floating chamber over the water. The *slope* was calculated with the variation curve of $pCO_2$. The emission pulses were excluded, and the slope was accepted only when the fitting curve had a $R^2$ higher than 0.90. Water temperature, pH, conductivity and dissolved oxygen (DO) were measured in situ with a portable multiparameter meter (Orion Star A321, Thermo Scientific, USA) with a resolution of 0.1°C, 0.01, 0.01µS/cm and 0.01mg/L, respectively. All the probes were calibrated before each sampling campaign started according to the manual. Due to malfunction of the instrument, the DO data was not available from September. Air temperature and wind velocity were measured with a portable anemometer (GM8901, Benetech, China). All the parameters were measured three times to reduce systematic error. For quality control, at least three water samples were collected from 0.5m below the water surface with water bottles. For alkalinity, the water samples were titrated with 2M hydrochloric acid within 12 hours after collection. The acid solution was titrated with NaOH solution. The data of alkalinity, pH and water temperature were used to calculate the $pCO_2$ of water samples with CO2SYS program (Lewis et al., 1998). The water samples were stored in 50ml centrifugal tubes and transported to the lab at a low temperature.

### 2.2.2 Laboratory analysis

The water samples for analysis of chlorophyll concentration were filtered with qualitative filter paper (80~120 µm), while the water samples for DOC analysis were filtered with 0.7µm Whatman GF/F filters to remove the sediments. Concentration of chlorophyll was analyzed with a Phyto-PAM-II Multiple Excitation Wavelength Phytoplankton analyzer (Heinz Walz GmbH, Germany). The DOC analysis was conducted on the Vario TOC Analyzer (Elementar, Germany). The resolutions of the analyser for chlorophyll and DOC were 0.01 µg/L and 0.001ppm, respectively. Unfiltered water samples were analyzed with spectrophotometer (UV5500, Metash, China) after digestion with alkaline potassium persulfate and potassium persulfate for concentration of total nitrogen (TN) and total phosphorous (TP) according to HJ636-2012 (MEP, 2012) and GB11893-89 (MEP, 1989), respectively.

# 3 Results

## 3.1 Spatial and temporal variation of environmental factors

Seasonal variation of temperature and rainfall reflected characteristics of monsoon climate (Fig.S1). In winter (from December to February), the air temperature was below 5 ºC, while the monthly average temperature was all over 25 ºC in summer (from June to August). The peak discharges of inflows in the mainstem and the tributary were both recorded in July, which were $70.50*10^8 m^3/s$ and $4.02*10^8 m^3/s$. Summer inflows accounted for 47% and 65% of the annual discharge in the mainstem and the tributary, respectively. The inflow water was characterized by low temperature, pH and TN concentration and high alkalinity, conductivity, DOC concentration, TP concentration and chlorophyll, while the pelagic zone was filled with warm, more alkaline, eutrophic, and less aerobic water (Table 1). The average water temperature of all the points ranged from 15.6 to 17.4 ºC, with an average of 16.8 ºC. The difference in water temperature between riverine zone and pelagic zone was no more than 2℃. Since the epilimnion water was used for hydropower generation, the water temperature at the downstream of the dam was very close to the surface water at the upstream of the dam. The pH values were mostly higher than 8.0 (8.46 in average), which suggested that the water in the reservoir was alkaline without any significant spatial heterogeneity. Total alkalinity ranged from 2251 µmol/L to 2666 µmol/L, with a mean value of 2441 µmol/L. Points located at the upstream had higher alkalinity than in the downstream pelagic area with the maximum recorded in the littoral zone. Ranging from 345 µS/cm to 388 µS/cm, conductivity showed a similar variation trend as alkalinity. The dissolved oxygen concentration in the pristine channel was approximately 4 mg/L, higher than that in the pelagic area. Concentration of DOC was also significantly higher in the riverine zone than in pelagic area, but it was quite homogeneous within the reservoir, possibly due to severe deposition. Both the concentration of TN and TP showed low values in the reservoir, with a mean value of 0.71 mg/L and 0.15 mg/L, respectively. The maximum concentration of nutrients was found in the littoral zones and pelagic area rather than in the riverine area in the mainstem.

## 3.2 Spatial and seasonal variation of pCO$_2$

Most of the water samples had pCO$_2$ higher than the atmospheric value (410 µatm) (Fig. 3), suggesting that the reservoir was a CO$_2$ source to the atmosphere. The pCO$_2$ of water samples ranged from 237 µatm to 14764 µatm, with an annual average of 919 µatm and a median of 711 µatm. The values were close to the global average of artificial reservoirs (Raymond et al., 2013).

Annual mean pCO$_2$ of the reservoir (703 ± 407 µatm) was comparable to the natural lakes in the Yunnan-Guizhou Plateau (639 µatm, Wang et al., 2003) when the pCO$_2$ from the river channel was excluded. The value was much lower than the pCO$_2$ in the Lower Mekong River (Li et al., 2013). Although there were no data available from the origin of the Mekong

River, the research on the three rivers on the Tibetan Plateau showed a median $pCO_2$ of 864 µatm, which was comparable to the values in the GGQ (Qu et al., 2017).

The $pCO_2$ was 852 ± 1056 µatm and 733 ± 232 µatm in the inflow of the mainstem and the tributary, respectively. These values were slightly higher than the $pCO_2$ in the surface water of the pelagic zone, but the difference was insignificant ($p>0.05$). Since the pH was higher than 8 with little variation, the $pCO_2$ showed no significant spatial heterogeneity in the reservoir in the spring, summer and winter. The $pCO_2$ was below 800 µatm from May to August, while it increased drastically in the late August. From September to April, the water level gradually rose and the $pCO_2$ fluctuated between 400 µatm and 1,200 µatm.

Variation of the $pCO_2$ was significant ($p<0.05$) among four seasons and the $pCO_2$ in autumn was much higher than in the other seasons. When the $pCO_2$ in the riverine area and the pelagic zone recorded their peak values in autumn, a significant decreasing trend toward downstream was found along the mainstem ($p<0.05$), which could be related to the low pH at the reach from R1 to L (Fig. 3). Frequent fluctuation of the water level and continued rainfalls flushed plenty of deadwood and organic matter to the reservoirs. Decomposition of the deadwood and plants could acidify the water along the bankside, which finally led to much higher $pCO_2$ in R1, P1 and L. Accumulation of deadwood was most obvious in the littoral zone. The $pCO_2$ in the littoral zone was 14764 µatm and 11825 µatm in September and October, respectively. The extremely high $pCO_2$ in the littoral zone indicated that this zone could be a potential "hotspot" for carbon emissions.

The $pCO_2$ measured at the downstream of the dam was quite stable throughout the year ($p>0.50$), with an average of 658 ± 176 µatm. No drastic increase from P3 to D was found throughout the year. The gradient in $pCO_2$ between P3, the point close to the dam, and D, at the downstream of the dam, ranged from -247 µatm to 560 µatm. The $pCO_2$ was found to be lower at the downstream of the dam than upstream from August to November. Unlike the cascade reservoirs on the Maotiao River where higher $pCO_2$ at the downstream of the dam had been recorded (Wang et al., 2011), the $pCO_2$ at the downstream of the GGQ rarely reached to 10,000 µatm.

### 3.3 Spatial and seasonal variation of $CO_2$ effluxes

$CO_2$ effluxes displayed large spatial and seasonal variation in the GGQ ($p<0.01$, Fig. 4 and Fig. 5). $CO_2$ effluxes ranged from -44 to 4952 mg $CO_2$ $m^{-2}$ $d^{-1}$ with a mean value of 352 ± 587 mg $CO_2$ $m^{-2}$ $d^{-1}$. One negative value was found in P4. It confirmed that the reservoir was a carbon source to the atmosphere, but the evasion rate was much lower than the estimated global average (Deemer et al., 2016; Holgerson & Raymond, 2016; Vincent et al., 2000). The annual effluxes at P1, P2, P3 and P4 were 465 ± 529mg $CO_2$ $m^{-2}$ $d^{-1}$, 331 ± 94 mg $CO_2$ $m^{-2}$ $d^{-1}$, 336 ± 92mg $CO_2$ $m^{-2}$ $d^{-1}$ and 273 ± 11mg $CO_2$ $m^{-2}$ $d^{-1}$,

respectively. Effluxes in the pelagic zone were lower in the summer and autumn than in the winter and spring, but the seasonal variation was not significant (p>0.50).

Fig. 5 displayed a decreasing trend of $CO_2$ efflux toward downstream. The annual efflux from the river channel was 1577 mg $CO_2$ m$^{-2}$ d$^{-1}$ and 905 mg $CO_2$ m$^{-2}$ d$^{-1}$ in the mainstem and the tributary, respectively, which was significantly higher than that in the reservoir area (p<0.50). The efflux in R1 was very sensitive to the monsoon climate. During the summer the efflux in R1 was no more than 274 mg $CO_2$ m$^{-2}$ d$^{-1}$, but it rapidly climbed to 2359 mg $CO_2$ m$^{-2}$ d$^{-1}$ at the end of October. The efflux stayed above 6,000 mg $CO_2$ m$^{-2}$ d$^{-1}$ in the winter and the high rate persisted till the following March. Hence, the difference in efflux between the river and the reservoir was more significant in the dry season than in the wet season.

The average efflux at Point D at the downstream of the dam was similar to that of Point P3 ($341 \pm 158$ mg $CO_2$ m$^{-2}$ d$^{-1}$), aligned with the results of $pCO_2$ (Fig. 3). The emission rate at the downstream was higher in the summer and winter but dropped below 300 mg $CO_2$ m$^{-2}$ d$^{-1}$ in the spring and oscillated between 200 and 300 mg $CO_2$ m$^{-2}$ d$^{-1}$ in the autumn. The low values contradicted the findings for many tropical reservoirs (Abril et al., 2005; Chanudet et al., 2011), but were consistent with the low $pCO_2$ reported for some mountainous reservoirs in eastern China (Zhao et al., 2013). The areal efflux downstream of the dam was consistently lower than that from the epilimnion in the reservoir because degassing could occur when the water passed through the turbine for electricity generation. It suggested that the carbon emission rates downstream of the dam were determined by the position of the water inlet and source layer of the water passing through the turbine.

The littoral zone had the highest emission rates within the reservoir ($684 \pm 1153$ mg $CO_2$ m$^{-2}$ d$^{-1}$), although this value was less than one third of the efflux estimated for the drawdown areas in temperate reservoirs (Aufdenkampe et al., 2011; Li et al., 2015). This was mainly because of the higher $pCO_2$; in autumn the littoral zone had the highest $pCO_2$ and the highest efflux along the reservoir when the water level frequently fluctuated.

### 3.4 Diurnal variation of $CO_2$ effluxes

In the GGQ water properties exhibited diurnal variations. The water temperature increased from 13:00 to 19:30 but kept decreasing after 22:00. As the air temperature kept decreasing throughout the sampling period, the water was heated before 24:00 and started to lose heat to the atmosphere afterwards. The alkalinity dropped from 15:00 to 19:30 and increased after 20:00. With a mean value of 2904 µg/L, alkalinity increased slightly in the nighttime. The conductivity varied little with the value ranging from 527.7 µS/cm to 540.8 µS/cm. The wind speed was higher in the daytime; the maximum (3.5m/s) was recorded at 16:30, while in the nighttime the sampling point was dominated by calm wind conditions.

We also observed a significant diel variation in $CO_2$ efflux ($p<0.01$). Before 8pm, the efflux was below 400 mg $CO_2$ m$^{-2}$ d$^{-1}$, but rose to above 450mg $CO_2$ m$^{-2}$ d$^{-1}$ after 0:30 at midnight. Efflux drastically oscillated from 9pm to 11pm between 69 mg $CO_2$ m$^{-2}$ d$^{-1}$ and 712 mg $CO_2$ m$^{-2}$ d$^{-1}$. As shown in Fig. 6, the $CO_2$ efflux was two times higher in the night (from 19:00 to 7:00: $495 \pm 178$ mg $CO_2$ m$^{-2}$ d$^{-1}$ in average) than in the daytime (from 7:00 to 19:00: $247 \pm 171$ mg $CO_2$ m$^{-2}$ d$^{-1}$ in average).

The trend was verified by the discontinuous efflux measurements (Fig. 7); the nocturnal $CO_2$ flux ($1012.29 \pm 1016.84$ mg $CO_2$ m$^{-2}$ d$^{-1}$) was higher than the daytime flux ($766.87 \pm 740.43$ mg $CO_2$ m$^{-2}$ d$^{-1}$). The efflux was negatively related to air temperature, wind speed and pH, but positively related to conductivity, alkalinity and $pCO_2$ ($N=40$, $p<0.01$). The significant relationship between $pCO_2$ and efflux revealed that fluctuation of $pCO_2$ could be an important reason for the diurnal variation of efflux because efflux was largely dependent on the $pCO_2$ gradient between atmosphere and surface water. The

$pCO_2$ was also higher at night than that in the daytime, although the difference was insignificant (Fig. 6, $p>0.50$). The insignificant variation of $pCO_2$ might be attributed to the alkaline environment in the GGQ. High pH ($8.22 \pm 0.06$) and its small variation kept the $pCO_2$ at a low level and limited the variability of $pCO_2$. Thus higher efflux and $pCO_2$ at night might be resulted  from dominated respiration in the surface water when light was unavailable for photosynthesis, which was also commonly found in other reservoirs (Liu et al., 2016a; Peng et al., 2012; Schelker et al., 2016).

**4 Discussion**

**4.1 Damming effect on carbon effluxes in the Upper Mekong River**

In this study, the $CO_2$ emission rates of the four-year old reservoir were low and comparable to those of natural lakes (Xing et al., 2005, Wang et al., 2003). Even in the river channel, the highest effluxes were close to the effluxes from temperate reservoirs (Huttunen et al., 2002) and much lower than those from tropical reservoirs (Abril et al., 2005; Fearnside, 1997;

Guérin et al., 2006). There were multiple reasons for the low carbon effluxes. First, the upper Mekong River drains through the Tibetan Plateau and flows within a narrow valley before it reaches the GGQ. Because of poor vegetation in the catchment and intense precipitation during the rainy season, the catchment could not sustain fertile soil or provide abundant organic carbon for decomposition even in the wet seasons. A shortage of substrates for mineralization limited the production of carbon dioxide.

Secondly, damming the river greatly extended the water retention time and the riverine ecosystem gradually evolved into a limnetic ecosystem (Thornton et al., 1990). The extended water retention time in the pelagic zone of reservoirs was suitable for the development of phytoplankton communities. When light and temperature were favourable, intense photosynthesis consumed the $CO_2$ dissolved in surface water and lowered the emission rates (Yu et al., 2009). In extreme cases like algae

bloom, the surface water tended to absorb $CO_2$ from the atmosphere (Pacheco et al., 2014b). Thus, the valley-type reservoir exhibited a decreasing trend from the river towards the dam in the $pCO_2$ and the outgassing rates (Liu et al., 2009; Liu et al.,

2014; Mei et al., 2011). Anthropogenic nutrient input could accelerate the process of eutrophication. With abundant nitrogen and phosphorous input from sewage, the outgassing rates could be decreased to a level as low as in natural lake or even turned negative (Guo et al., 2011; Ran et al., 2011). The effluxes from the GGQ displayed a negative relation with the water temperature ($p<0.01$, Fig. 8). The negative relation deviated from the traditional pattern where a warmer climate accelerated bacterial respiration (Åberg et al., 2010; Del Giorgio & Williams, 2005) and decreased the solubility of carbon dioxide, thus enhancing the effluxes. This deviation suggested that warmer climate could also reduce the $CO_2$ emissions via accelerated photosynthesis.

The seasonal variation of efflux in the pelagic area, however, was less significant ($p>0.05$) than the variation in the riverine sites of Point R1 and R2 ($p<0.01$). The riverine inlets of the reservoir were identified as a hotspot of $CO_2$ emission in the dry season (from November to April), where the extremely high emission rates distinguished from pelagic area ($p<0.01$). In some large valley-type reservoirs rainfalls brought plenty of organic carbon and increased flow velocity, fuelling $CO_2$ emissions at the mainstem channels in the wet season (Li & Zhang, 2014; Zhao et al., 2013). Yet in this case the effluxes at the riverine points were negatively related to the water discharge (Fig. 9), water temperature, and nutrient concentration (Table S1), suggesting that higher emissions could happen at a lower flow velocity and a colder condition (Fig. 8&9).

This abnormal results could be explained by different mixing modes occurring at the riverine points when the inflow joined the reservoir, which could be represented by the differences in physical properties like temperature and turbidity (Summerfield, 1991). As shown in Fig. 10, the inlets had higher effluxes when the inflow water was warmer and contained less suspended sediments than the receiving waterbody. It suggested that the seasonal variation of effluxes was regulated by both flow mixing modes and reservoir management (Striegl & Michmerhuizen, 1998). Even though in the rainy season intense precipitation could bring plenty of sediments with organic matter, the turbid water might be discharged directly to the downstream for electricity, because of the relatively small storage capacity of the reservoir. The inflow water with high sediment concentration was heavier and colder than the reservoir water, thus it plunged  into the water column in the reservoir and became an underflow (hyperpycnal flow) (Fig. 10) (Summerfield, 1991). The reservoir surface was less affected by the underflow and maintained a relatively low emission rate (Pacheco et al., 2014a) as continuous water discharging allowed little time for the mineralization of organic carbon (Assireu et al., 2011; Senturk, 1994), in spite of the high flow velocity. However, in the dry season the clean inflow water was lighter and warmer than the reservoir water, and thus it joined the reservoir as surface flow (hypopycnal flow) (Fig. 10) (Summerfield, 1991). The data in Fig. 3 showed that the inflow water in the winter (the dry season) was also richer in $CO_2$ than the turbid inflow in the summer (the wet season). When the water rich in $CO_2$ contacted the atmosphere directly, the gases directly diffused into the air. Because the water kept losing $CO_2$ to the atmosphere, the decreasing trend in effluxes towards downstream was more significant in the winter (Fig. 5).

Due to this difference in physical mixing modes and availability of $CO_2$, the surface water tended to release more $CO_2$ in the dry season when both inflow and reservoir water became colder (Fig. 4). It was likely that the underflow in the rainy season also mixed and aerated the water in the reservoir and thus impeded the formation of stratification. The efflux in the downstream was restricted and showed a similar seasonal variation to the reservoir surface water. During stratification, the downstream river channel could have released substantial $CO_2$ if the water from hypolimnion was used to generate electricity (Abril et al., 2005; Guérin et al., 2006; Wang et al., 2011).

The littoral zone (or drawdown area) displayed much higher effluxes than the pelagic zone, especially in the autumn and winter. The littoral zone had often been identified as a hotspot of carbon emission (Chen et al., 2009; Yang et al., 2012; Yang, 2011) since seasonal flooding could trigger anaerobic decomposition of dead macrophytes and produced greenhouse gases. In this case, it was believed that the frequent fluctuation of water level deposited a large amount of sediments as well as deadwood on the relatively flat littoral zones. The decomposition of deadwood tended to release organic acids to the water and lowered the pH. As a result, the $pCO_2$ rose and more gases were degassed out of the air-water interface. Furthermore, nutrient inputs and reduced turbidity facilitated growth of plants and macrophytes (Thornton et al., 1990) and enhanced respiration and $CO_2$ outgassing (Xu, 2013) (Fig. S2).

## 4.2 Extrapolation of the results and implication for future studies

The efflux from the pelagic zone and from the littoral zone was $352 \pm 587$ mg $CO_2$ m$^{-2}$ d$^{-1}$ and $684 \pm 1153$ mg $CO_2$ m$^{-2}$ d$^{-1}$, respectively. Assuming the water level fluctuated frequently within 2.5 m and the slope at the bank was 45°, the drawdown area covered an area of $1.81 \times 10^5$ m$^2$. Hence the littoral zone could contribute $22.59 \pm 38.09$ t $CO_2$ yr$^{-1}$ to the atmosphere, assuming it would be flooded in half of the year. We estimated that the permanent flooded area was 5,643,000 m$^2$ in the GGQ and the carbon dioxide evading from this area was $725.01 \pm 1209.04$ t $CO_2$ yr$^{-1}$. Compared with the estimated emission, the contribution from the littoral zone was actually negligible for its small area. However, if taking the diurnal variation into account, the annual carbon evasion reached to $1121.41 \pm 1209.64$ t $CO_2$ yr$^{-1}$ as nocturnal effluxes was twice as the emission in the daytime. Considering its efficiency, the reservoir released $0.28 \pm 0.30$ kg $CO_2$ per MW/h when generating hydroelectricity. This estimation was close to the lower bound of the range (0.2~1994 kg $CO_2$ per MW/h) estimated by Räsänen et al. (2018). However, it must be noted that the $CO_2$ efflux would decrease as the reservoir ages (Abril et al., 2005; Barros et al., 2011). Accelerated eutrophication could possibly fix more $CO_2$ via photosynthesis (Liu et al., 2009).

Several problems have been noticed when computing the annual emission rate from the GGQ. Despite its higher efflux, the drawdown area was negligible although the effluxes from global reservoirs always displayed high spatial heterogeneity (Barros et al., 2011; Roland et al., 2010; Teodoru et al., 2011). On a larger scale the seasonal variation was also negligible as

the efflux in the dry season was only 103 mg $CO_2$ m$^{-2}$ d$^{-1}$ higher than in the rainy season. However, the higher effluxes in the nighttime must be taken into consideration. Measurement of the effluxes from the reservoir surface was usually limited by the $pCO_2$ samples collected in the daytime and failed to capture a diurnal variation, though this variation has been fully recognized by a series of studies (Liu et al., 2016a; Peng et al., 2012; Schelker et al., 2016).

The sediment deposition must also be considered when computing the long-term effect of reservoir on carbon cycle. As the uppermost reservoir along the Lancang cascades, the GGQ also sequestered most of the sediments from the upstream catchments (Gao et al., 2017; Wang et al., 2011). It is likely that the reservoir cannot be maintained for 100 years due to heavy silting problem (Fu & He, 2007), even though the sediment concentration has decreased drastically after the upstream

Miaowei Dam was completed, enabling the reservoir to bury tons of organic carbon (Mendonça et al., 2012b; Mulholland & Elwood, 1982; Vörösmarty et al., 2003). Meanwhile, the reservoirs could also sequester the nutrients in the rivers (Maavara et al., 2017; Maavara et al., 2015). Therefore, in order to evaluate the net effect of impoundments on carbon cycle, we need to quantify the organic carbon burial within the reservoir and finally build up a robust carbon budget.

**5 Conclusion**

The surface water of the GGQ was supersaturated with $CO_2$ and the reservoir was a carbon source to the atmosphere. We estimated that the reservoir released 1121.41 $\pm$ 1209.64 tons of $CO_2$ to the atmosphere annually. The efflux from reservoir area was 408 $\pm$ 337 mg $CO_2$ m$^{-2}$ d$^{-1}$ and 305 $\pm$ 262 mg $CO_2$ m$^{-2}$ d$^{-1}$ in the dry season and rainy season, respectively, while the river channel exhibited an efflux of 2168 $\pm$ 2547 mg $CO_2$ m$^{-2}$ d$^{-1}$ and 374 $\pm$ 184 mg $CO_2$ m$^{-2}$ d$^{-1}$ in the two seasons. The $CO_2$ emission from the pelagic zone was limited due to little allochthonous organic carbon input and photosynthetic uptake.

Seasonal variation of efflux in the reservoir was subject to the variation of temperature, with lower emission rates occurring in the warmer wet season (May to October) owing to enhanced photosynthesis. Emissions at downstream of the dam was also limited as surface water was used for generating electricity. However, the littoral zone suffering frequent flooding and draining was identified as a potential hotspot of $CO_2$ emissions, even though its contribution to the total annual emission was limited due to its small area. Flat topography and daily flooding could lead to accumulation of deadwood and acidification of

water, aerate the water and enhance the respiration rate.

This study also highlighted the high emission rates at the river inlets during the colder dry season. The negative relation between efflux and water discharge implied that the mixing modes could be the dominant factor controlling $CO_2$ emissions. In the winter, because inflow was warmer, clearer and lighter than the receiving waterbody, the gas carried by inflow could

be more easily released to the atmosphere as the river joined the reservoir as an overflow. Additionally, extended water retention time was also beneficial for decomposition of allochthonous DOC and produced more carbon dioxide. In the wet

season, when the inflow plunged into the reservoir, the unferflow could be discharged directly to the downstream and left insufficient time for the mineralization of DOC. The physical factors could be an important factor controlling the $CO_2$ emissions beside the biological factors for hydroelectric reservoirs where the hydrological conditions were regulated by climate and artificial operation. Yet in a daily cycle, the biological factor could cause significant diel variation, as emissions could be offset by the carbon absorption via photosynthesis. The total emission from the GGQ increased by half when taking the nocturnal effluxes into account. Hence, the efflux measured in the daytime must be carefully integrated when estimating the total carbon emissions from a reservoir. In this study, the damming effect on the CO2 emission from waterbody was moderate, but for an overall effect on carbon transportation a robust carbon budget was required in which the carbon burial in sediments must also be quantified.

## Acknowledgements

The research reported here has received funding from the National Natural Science Foundation of China (Grant No 91547110; 41571032) and financial support from National University of Singapore (Grant No. R-109-000-191-646; R-109-000-227-115).

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

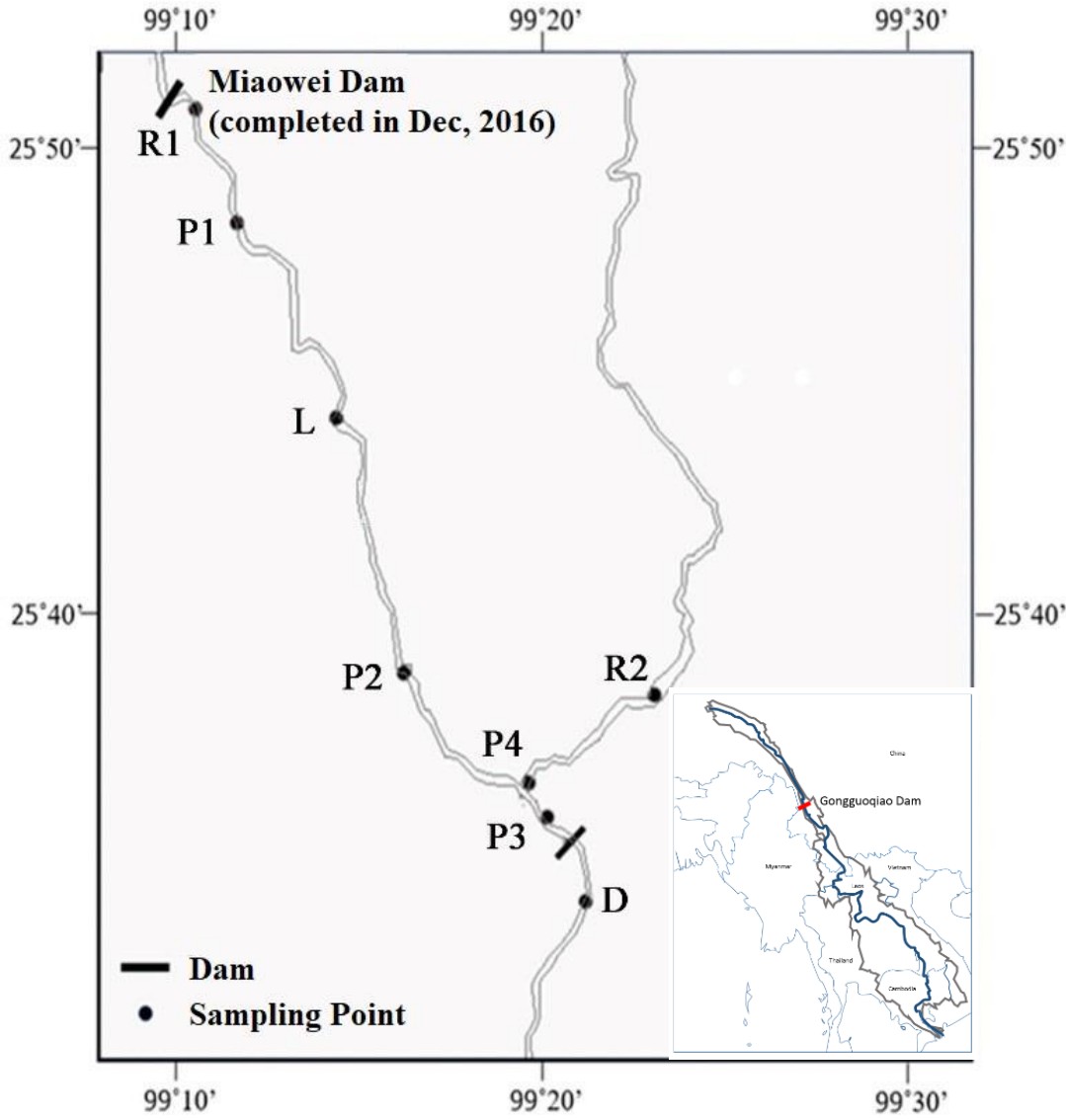

**Figure 1 Sampling points in the Gongguoqiao Reservoir and its position within the Mekong River Basin. Point L1 is downstream the Miaowei Dam which was completed in Dec, 2016. Point R1 and R2 was in the river channel with flow velocity. Point P1, P2, P3 and P4 were in the reservoir without flow velocity. Point D was at the downstream of the reservoir. Point R2 and P4 were in the tributary the Bijiang River while all the other points were in the mainstemof Mekong River (or Lancang River).**

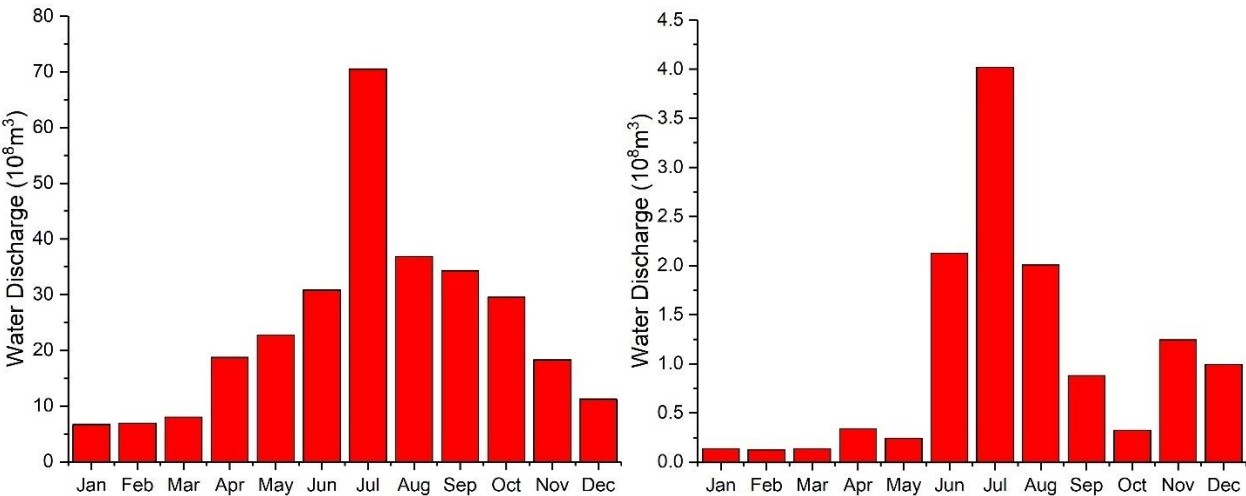

**Figure 2 Monthly water discharge of the inflow at the mainstem (left panel) and the tributary (the Bijiang River, right panel) into the GGQ Reservoir. Notice that the inflow from the tributary was estimated with the instant water discharge (m3/s). The instant water discharge was measured at the same time as the sampling campaign at Point R2 at the Lanping or Yunlong Hydrological Gauging Station, which was about 30km away from Point R2.**

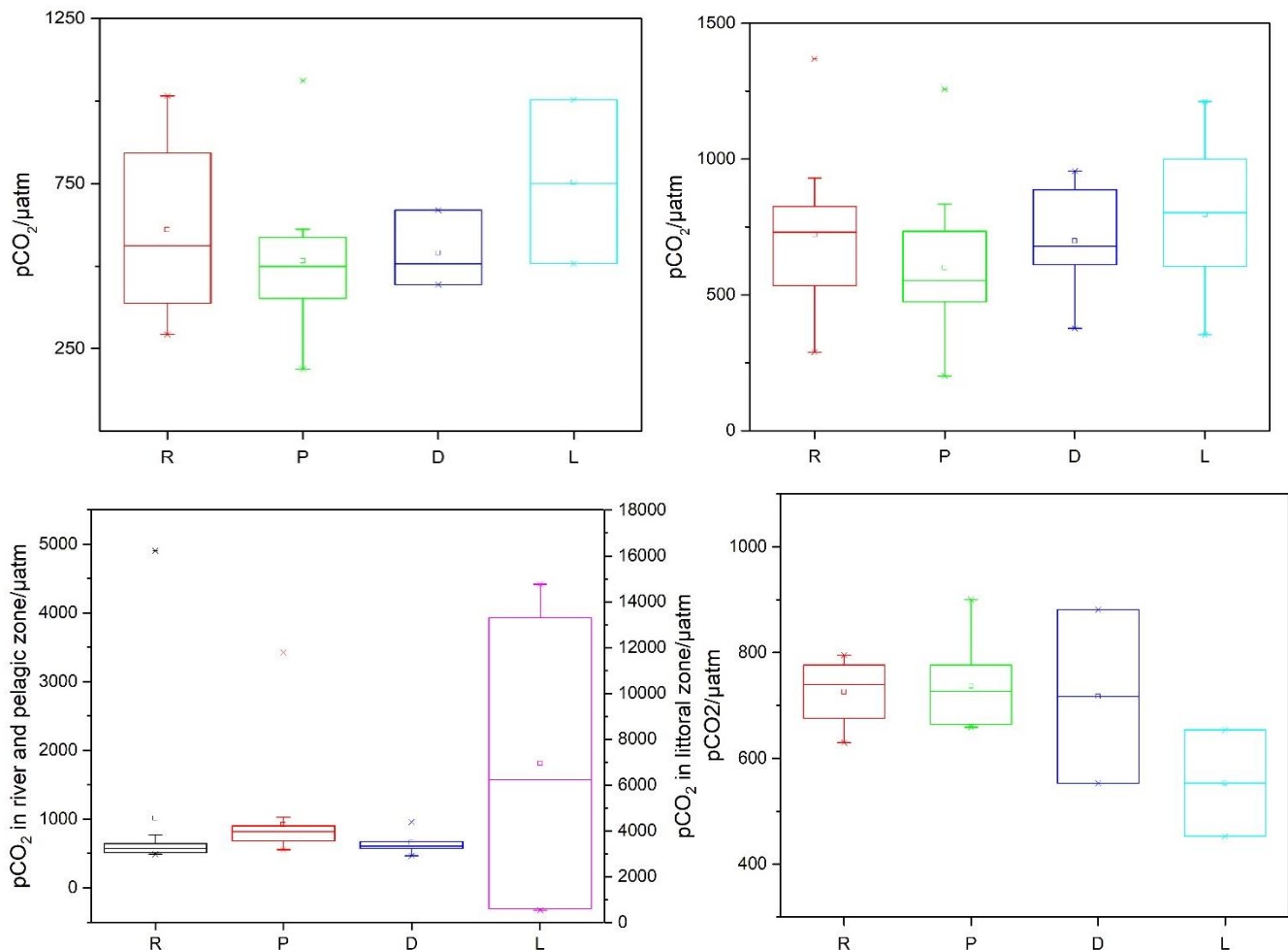

**Figure 3 Box Plots of pCO₂ in the rivers (R), permanent flooded area of the reservoir (P), downstream (D) and littoral zone (L) in the spring (upper left panel), summer (upper right panel), autumn (lower left panel) and winter (lower right panel). Notice that the scale of pCO₂ at the littoral zone in autumn was shown on the scale of right hand side. The lower and upper bound of the box refers to the first and third quartile respectively and the vertical line indicates the 1.5 interquartile range. The points outside the range was considered outliers and are represented by little cross. Horizontal line refers to the median value while the little squares refers to the average.**

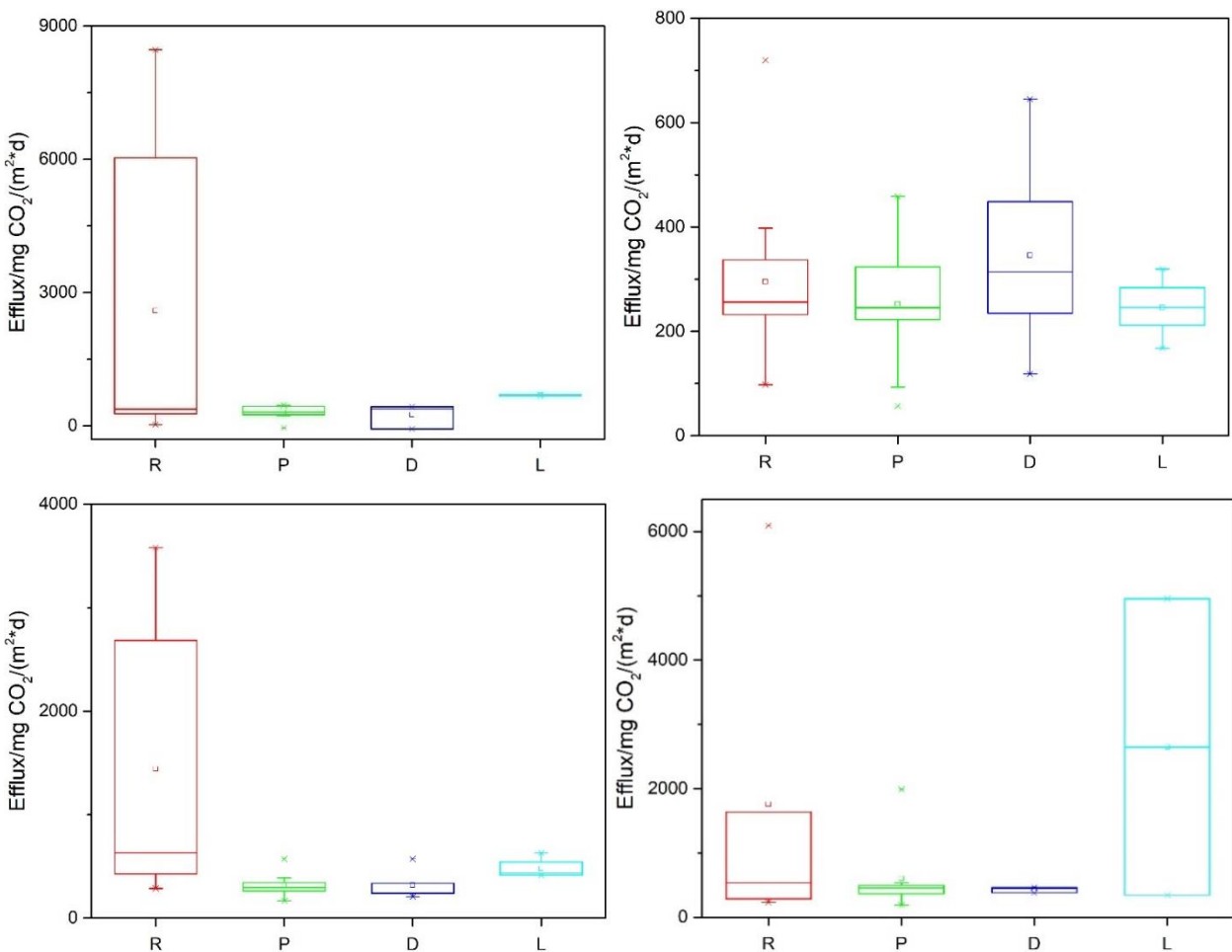

**Figure 4 Box plots of the measured CO₂ effluxes in the spring (upper left panel), summer (upper right panel), autumn (lower left panel) and winter (lower right panel). The lower and upper bound of the box refers to the first and third quartile respectively and the vertical line indicates the 1.5 interquartile range. The points outside the range was considered outliers and are represented by little cross. Horizontal line refers to the median value while the little squares refers to the average.**

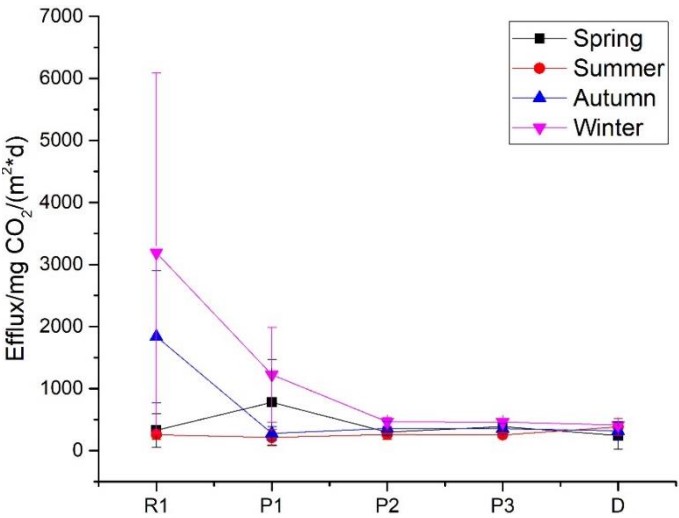

**Figure 5 Longitudinal variation in effluxes along the mainstem in different seasons. The points and error bar refer to mean value and standard deviation respectively.**

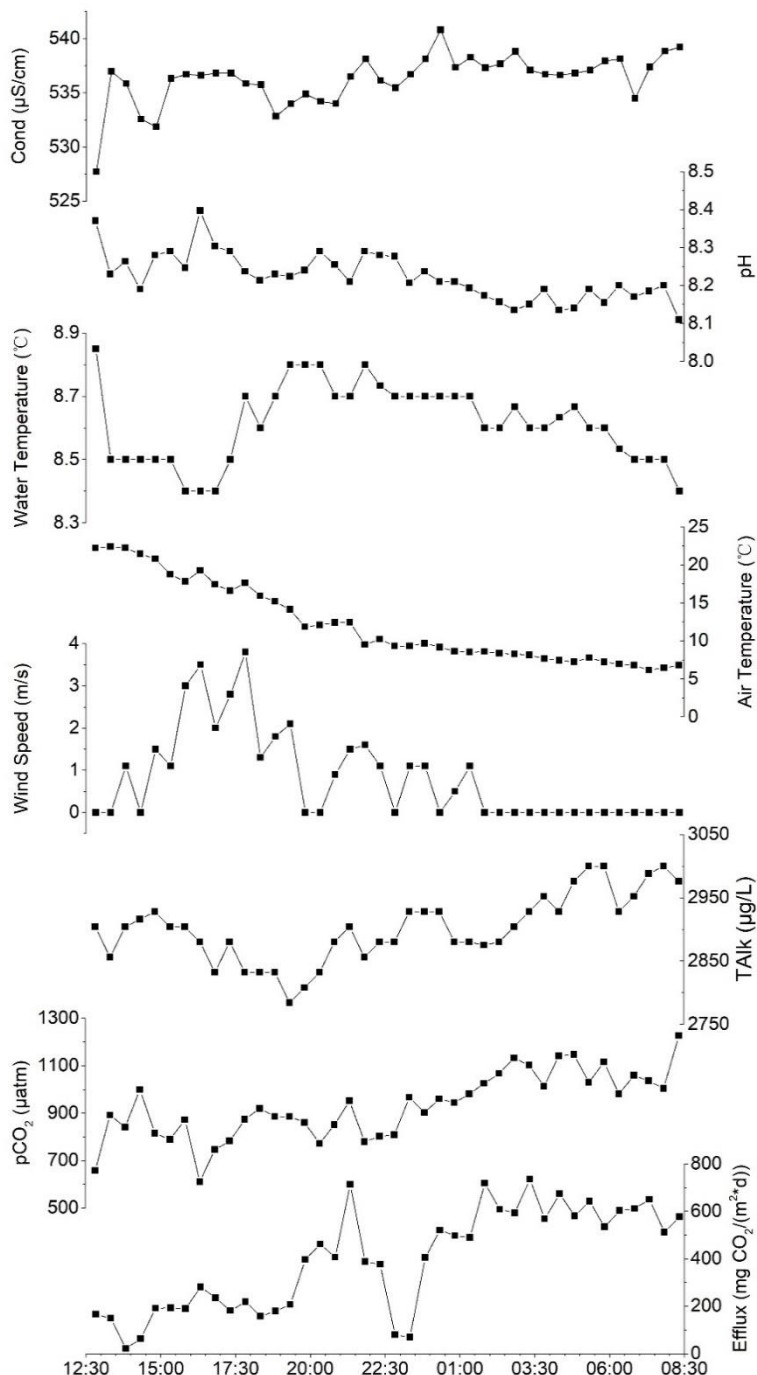

**Figure6 Diurnal variation of the water environment (including conductivity, pH, water temperature and total alkalinity), atmospheric environment (air temperature and wind speed) $pCO_2$ and efflux.**

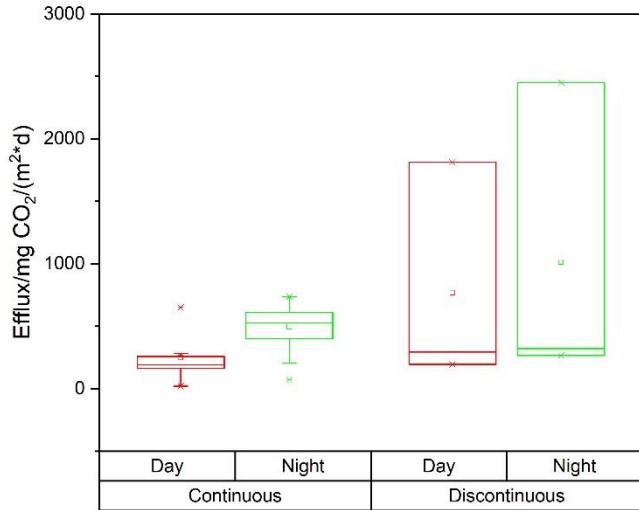

**Figure7 Comparison in effluxes between daytime and night via continuous samples (left panel) and discontinuous samples (right panel).  The lower and upper bound of the box refers to the first and third quartile respectively and the vertical line indicates the 1.5 interquartile range. The points outside the range was considered outliers and are represented by little cross. Horizontal line refers to the median value while the little squares refers to the average.**

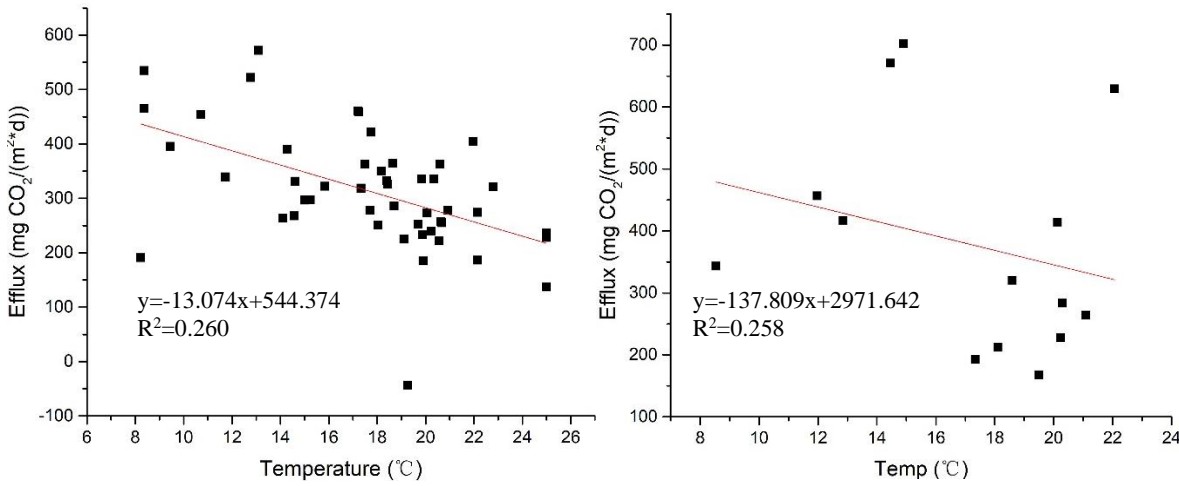

**Figure 8 Negative correlations between water temperature and effluxes in the pelagic zone (left, p<0.01) and in the littoral zone (lower right, p>0.05). Notice that two extreme values were excluded out in the linear regression in the upper right panel**

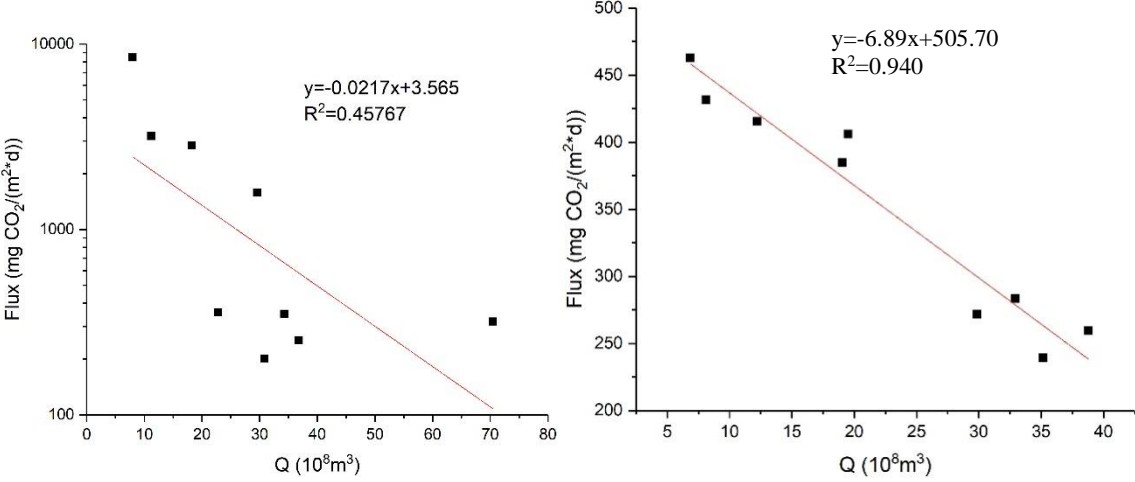

**Figure 9 The negative correlation between water discharge and $CO_2$ efflux at the riverine inlet (R1, left panel, p<0.01) and outlet (D, right panel, p<0.01)**

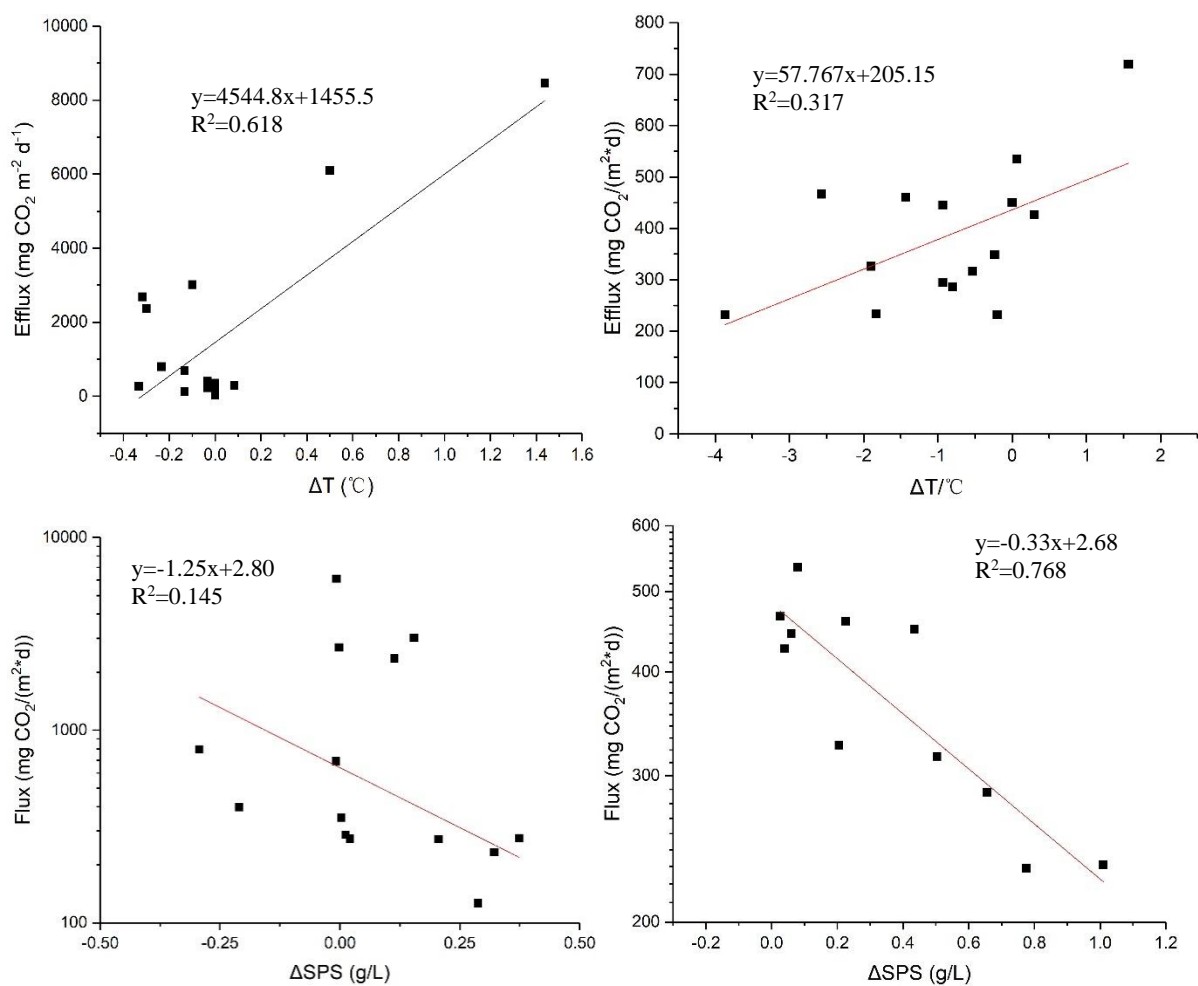

**Figure 10 Positive correlations between water temperature gradient (TR1-P1 or TR2-P4) and measured effluxes at R1 (upper left, p<0.01) and R2 (upper right, p<0.05), and the negative correlations between SPS concentration gradient (SPS R1-P1 or SPS R2-P4) and measured effluxes at R1 (lower left, p<0.01) and R2 (lower right, p<0.01). The gradient in water temperature and SPS concentration reflects the difference of properties between inflow and receiving waterbody and determines the mixing mode. Colder and more turbid inflow has higher density than the receiving water and thus forms an underflow or subsurface flow (hyperpycnal flow). When the inflow was warmer clearer and lighter than the receiving waterbody, the inflow can form a surface flow (hypopycnal flow) and flow over the reservoir surface, releasing allochthonous carbon to the atmosphere.**

**Table 1** Median temperature (Temp), pH, total alkalinity (Talk), conductivity (Cond), dissolved oxygen (DO), concentration of dissolved organic carbon (DOC), concentration of chlorophyll a (Chl a), concentration of total nitrogen (TN) and total phosphorous (TP) of sampling points

| | Temp/°C | pH | Cond/μS/cm | DO/mg/L | Talk/μg/L | TN/mg/L | TP/mg/L | Chl a/μg/L | DOC/ppm |
|---|---|---|---|---|---|---|---|---|---|
| | Med (Min-Max) | Med (Min-Max) | Med (Min-Max) | Med (Min-Max) | Med (Min-Max) | Med (Min-Max) | Med (Min-Max) | Med (Min-Max) | Med (Min-Max) |
| R1 | 16.9(8.4-20.5) | 8.40(7.47-8.61) | 355.4(296.2-536.4) | 8.93 (8.08-19.33) | 2608(1696-3036) | 0.51 (0.04-1.40) | 0.12 (0.01-0.73) | 0.99 (0.73-2.34) | 2.78 (2.48-5.38) |
| R2 | 19.2(8.3-21.1) | 8.35(8.09-8.80) | 295.0(159.8-437.7) | 7.97 (4.61-20.16) | 2508(1888-3456) | 0.69 (0.20-4.47) | 0.30 (0.01-1.65) | 1.15 (0.75-2.09) | 2.47 (2.47-4.82) |
| P1 | 17.1(8.3-20.5) | 8.38(7.63-8.86) | 352.5(256.6-540.4) | 8.81 (8.03-10.05) | 2486(1712-2928) | 0.51 (0.04-1.66) | 0.04 (0.01-0.65) | 1.01 (0.61-2.68) | 2.88 (2.42-6.42) |
| P2 | 17.8(8.4-25.0) | 8.35(8.03-8.84) | 330.5(214.2-537.2) | 8.66 (7.94-9.32) | 2338(1528-2928) | 0.59 (0.04-2.30) | 0.02 (0.01-0.52) | 0.92 (0.75-1.68) | 2.64 (2.60-5.74) |
| P3 | 18.6(8.4-25.0) | 8.28(8.05-8.49) | 333.0(253.2-462.9) | 8.30(7.49-8.83) | 2262(1800-2772) | 0.65 (0.04-1.59) | 0.02 (0.01-0.49) | 0.95 (0.62-1.84) | 2.62 (2.57-3.77) |
| P4 | 19.6(8.2-25.0) | 8.34(8.08-8.77) | 343.6 (259.4-494.2) | 7.90(7.63-9.87) | 2220(1888-2928) | 0.79 (0.04-2.78) | 0.02 (0.01-0.12) | 0.99 (0.61-1.18) | 2.67 (2.73-5.34) |
| D | 17.5(8.3-25.0) | 8.37(8.17-8.62) | 340.1 (266.0-529.2) | 9.90(7.96-20.11) | 2508(1784-3000) | 0.52 (0.03-1.88) | 0.02 (0.01-0.71) | 0.99 (0.63-2.05) | 2.77 (2.56-5.49) |
| L | 18.1(8.5-22.1) | 8.34(7.00-8.53) | 357.7(275.4-539.4) | 8.49(6.77-9.07) | 2736(1928-4320) | 0.61 (0.04-2.48) | 0.02 (0.01-0.50) | 0.98 (0.63-1.60) | 2.81 (2.26-4.02) |

