# Peer review of "Physical-controlled CO2 effluxes from reservoir surface in the upper Mekong River Basin: a case study in the Gongguoqiao Reservoir"

_Biogeosciences, 2018_

## Referee Comment (RC1) · Anonymous Referee #1 · 13 Aug 2018

This paper describes CO2 concentration and flux measurements made over a ∼1 year period upstream, downstream, and within profundal and littoral regions of a run-of-river (short residence time) reservoir in the Mekong River Basin. While the role of reservoirs in carbon and greenhouse gas budgets is an important and timely topic, I think this paper needs significant restructuring and re-framing before publication in Biogeosciences. Firstly, I don't think the argument that few CO2 efflux measurements have been made in China is substantiated (see global map in Figure 2 of Deemer et al. 2016). The authors even cite a number of other studies of carbon dioxide dynamics in Chinese reservoirs. I think the authors could emphasize the importance of understanding these dynamics in the Mekong basin given all the reservoir development that

is slated for the region (maybe cite Zarfl et al. 2015 Aquatic Sciences). The authors could also do a better job of describing the unique hydrology/climate in the Mekong Basin since the diverse readership base may not be familiar with the characteristics of dry vs. wet seasons in this region. Secondly, I think the authors should be careful in their discussion of global carbon budgets vs. reservoirs as greenhouse gas emitters—specifically, there is no mention in the paper about the potential role of methane as a GHG source and it is somewhat implied that CO2 might be the dominant emission pathway even though it is generally accepted that methane is often the dominant GHG source on an CO2 equivalent basis. Thirdly, I think the authors need to better integrate the diel sampling component of their study into the way that the other results are analyzed. The authors don't mention the temporal sampling scheme employed during their 16 sampling campaigns—were sites always sampled in the same order? Over what range of times? Are we confident that variation in fluxes measured is more a function of spatial variation than temporal variation? Fourth, while I think that hydrology may be a dominant control on reservoir CO2 emissions in this reservoir (e.g. it seems a completely valid and plausible hypothesis), I don't think the authors present enough evidence in support of this mechanism to present it as a result (e.g. in the abstract of the paper). Reservoir hydrology co-varies with other seasonal variation in temperature and the authors present no systematic approach for differentiating other possible controls. Finally, it is difficult to interpret the zonation grouping—the authors should consider incorporating a statistical assessment of significant differences between sites. For example, were the riverine samples from both sites more similar to each other than to other sites? Or was one riverine site emitting CO2 at much higher rates than the other? Reservoir inlets are often hot spots for biogeochemical activity—are we sure that these riverine sites are fully riverine and that their hydrology isn't influenced by the dam? In addition to these scientific concerns, the manuscript needs to be edited for proper English. There are grammatical issues and vaguely written statements that could benefit from a third-party editor.

Line By Line Edits Page 1 Line 12: change "cycle" to "cycling" Page 1 Line 14: did

the authors use a statistical approach to see if reservoir emissions were significantly different by season? Page 1 Line 17: I don't think the analysis presented here conclusively linked $CO_2$ emissions to physical mixing. Page 2 Lines 3-5: Carbon dioxide is generally thought of as the largest contributor to total carbon emissions, but methane is generally the largest contributor to total greenhouse gas emissions on a $CO_2$ equivalent basis. I think the authors should be careful to make this distinction clear. Page 2 Line 18: By "biogeochemical processes of phytoplankton" do you just mean photosynthetic uptake? Page 2 Line 24: The way you have phrased this sentence makes it sound like all the studies you are citing were conducted in the Three Gorges Reservoir, but Pacheco et al. 2014 was in Brazil. Also, I don't see Tao 2017 listed in your references section. Page 2 Line 25: Do you mean watershed? Not waterbody? Page 3 Line 5: Why is information about Xiaowan Reservoir relevant here? Also, perhaps this is a good place to mention the construction of Miaowei Dam (which is noted in your Figure 1). Was the dam completed after your sampling ended in Dec 2016? Was the system hydrology affected at all by the fact that a dam was being constructed upstream during your study? Page 3 Line 9: Is this a hydropeaking (load following) reservoir? It might be nice to see water level data from the reservoir given the current discussion of water level fluctuation you have incorporated into your discussion. Page 3 Line 16 (and throughout): You use "mainstream" when I think you mean "mainstem". Page 4 Line 2: Consider reformulating the equation to take out unit conversion factors (which seem a little distracting and un-necessary). Page 4 lines 26-28: The authors discuss dam hydrology as if they don't know what type of spill practices are employed in the reservoir. Isn't this information available? Also, the height of reservoir spill (epilimnion versus hypolimnion) could be mentioned in the study area section. Page 5 line 3: Why do the authors feel that the dataset is limited? Is there reason to think that sometimes the running waters from inflow are not more aerobic than the reservoir water? Page 5 line 11: Change this sentence to something like "With the exception of one sample, the reservoir was consistently supersaturated with $CO_2$, indicating its role as a $CO_2$ source to the atmosphere" Page 5 Lines 20-24:

A plot that shows water level and point $CO_2$ measurements over time might be helpful here—I got a little lost in this description of the results. Page 7 Line 1-2: Where do the authors show this analysis? Right now there is no mention of a statistical analysis of drivers and no corresponding table or figure. Page 7 Lines 4-21: So, given these results, are you confident that the $CO_2$ efflux measurements you made are still predominantly representing spatial (rather than temporal) variation? Also, it sounds like physical differences (rather than biology) may be driving the differential emissions you see during the day versus at night? Would you agree? Page 7 Line 24: How do you define a pristine river channel? Was R1 at all influenced by the construction of Miaowei Dam? How do you differentiate free-flowing river from reservoir inlet? Page 8 Lines 11-12: I don't think Figure 7 really shows this. Page 9 Line 8: Not sure "constraint" is the right word. Page 10 Line 17: Why "potential"? Page 10, Conclusion: No discussion about why emissions were so high from the river in the dry season. Was this pattern consistent in both river sites? Page 10, Line 31: What pattern are you referring to? Figure 6: Continuous versus incontinuous diel sampling was not explained in the methods.

Please also note the supplement to this comment:
https://www.biogeosciences-discuss.net/bg-2018-244/bg-2018-244-RC1-supplement.pdf

---

## Referee Comment (RC2) · Anonymous Referee #2 · 8 Sep 2018

This study reports CO2 emissions from an hydroelectric reservoir located in the upper Mekong river basin

The writing style makes difficult the review of this article.

The site description is incomplete:

-What was the land cover before flooding?

-What are the water discharge in and out of the reservoir according to seasons? –What is the average water residence time?

-Importantly, the seasons must be described precisely and the same nomenclature

must be used throughout the article instead of using sometimes summer, winter, spring. . . and at other places warm season, rainy season and even some combination like warm dry seasons. . . The reader is lost. . .

-Meteorological information like temperature and rainfall range are required

-the map (figure 1) requires a scale, an orientation and information about direction of the water flow would be welcome.

-is the reservoir thermally stratified? Well mixed? Monomictic?... Such information is required to be able to understand the seasonal dynamic of a lake or a reservoir

The sampling strategy requires clarification

-Can we call the station L as a littoral station since it seems to be an artificial island which has developed after sedimentation in the reservoir? In some part of the manuscript it is also called the drawdown area. . . Again, the reader is lost by the inconsistence of the vocabulary.

-P3-L19 stations P1 to P4 are considered all together whereas a few lines below, only P2-P4 are considered as pelagic stations. What type of station P1 is representative for?

-Not clear in the sampling strategy and site description but the sampling occurred during the year 2016 (P2L23) while the dam upstream of the study site was completed by December 2016 (Figure 1). Therefore, all the sampling might have been done during the construction which means that the river was heavily disturbed. The construction might have biased the conclusion on the fact that the "pristine river" (as the authors call it) emits more than the reservoir itself.

The methodology is minimalist and substantial information is missing to be able to evaluate the quality of the dataset:

-How many samples were gathered in total? By campaigns? Was the sampling organized by seasons?

-P3-L25-30: what are the precision, range and accuracy of the gas analyser? What gas flow was used? Did the author used desiccant? Is there a humidity correction is the analyser? What is the volume of the chamber? How were measured the fluxes in the river? At fixed station or drifting with the flow? What was the rejection/acceptance procedure for the measured fluxes?

-What are the precision and accuracy for Temp, O2, pH, conductivity measurements? This is critical for pH since pCO2 was calculated by pH/Alka method. Details on pH measurements are required

- Precision and accuracy and detection limit are required for Alkalinity.

- pCO2 using pH, Alkalinity and the CO2SYS program. This validity of the methodology was discussed recently by Abril et al. (2015) and (Golub et al., 2017) for inland waters

-For chlorophyll: How long after sampling the water was filtered? Were the filters kept in the freezer? What was the precision, accuracy and limit of detection for Chlorophyll, DOC, TOC, TN and TP?

-statistics used for the seasonal and spatial variations were not described

-the thin boundary method which was used according to P6L10 was not described

According to the fact that the sampling strategy and the validity of the pCO2 dataset is doubtful and the quality of dataset cannot be evaluated in absence of information, it is impossible to go further with the review of this manuscript.

Abril G, Bouillon S, Darchambeau F et al. (2015) Technical Note: Large overestimation of pCO2 calculated from pH and alkalinity in acidic, organic-rich freshwaters. Biogeosciences, 12, 67-78.

Golub M, Desai AR, Mckinley GA, Remucal CK, Stanley EH (2017) Large Uncertainty in Estimating pCO2 From Carbonate Equilibria in Lakes. Journal of Geophysical Research: Biogeosciences, 122, 2909-2924.

---

## Author Response (AR1)

**Response to the Reviews**

**#1**

Comment 1: Firstly, I don't think the argument that few CO2 efflux measurements have been made in China is substantiated (see global map in Figure2 of Deemer et al. 2016). The authors even cite a number of other studies of carbon dioxide dynamics in Chinese reservoirs. I think the authors could emphasize the importance of understanding these dynamics in the Mekong basin given all the reservoir development that is slated for the region (maybe cite Zarfl et al. 2015 Aquatic Sciences). The authors could also do a better job of describing the unique hydrology/climate in the Mekong Basin since the diverse readership base may not be familiar with the characteristics of dry vs. wet seasons in this region.

Response: Thanks for the reasonable comments on the citation. It was true that many studies on pCO2 in reservoirs accumulated in China. But many current CO2 efflux from reservoirs in China were estimated with pCO2 and wind speed but not direct measurements. Even though under most of the circumstances the pCO2 could predict the efflux effectively, we thought the CO2 efflux estimated in this way could be underestimated as some physical controls on CO2 emissions could be neglected. Hence, this research was based on quantification of the CO2 emissions from a reservoir in the upper Mekong River, because Most of the studies focus on the variation of pCO2 in surface water rather than the emissions.

Yet we agreed that the Introduction should be better to emphasize the importance of understanding these dynamics in the Mekong Basin. Thus, we will add some information about the South Asia monsoon climate and potential effect of artificial operation on the CO2 production and emissions from the reservoir and describe the potential monsoonal/hydrological effect on the CO2 emissions for those readers who do not familiar with the catchment. We hope that this could distinguish our study from other existing research on the dynamics of CO2 production and emphasize its necessity.

Comment 2: Secondly, I think the authors should be careful in their discussion of global carbon budgets vs. reservoirs as greenhouse gas emitters-specifically, there is no mention in the paper about the potential role of methane as a GHG source and it is somewhat implied that CO2 might be the dominant emission pathway even though it is generally accepted that methane is often the dominant GHG source on an CO2 equivalent basis.

Response: Yes, we agreed with the referee that the methane is the dominant GHG source on the CO2 equivalent basis at global scale. We were trying to say that quantitively the amount of carbon dioxide released from the reservoirs was higher than that of methane if the global warming potential was not considered. But the expression could be awkward and lead to misleading implication.

As suggested, we should be more careful in evaluating the effect of carbon dioxide vs. methane. Since the article focuses on the CO2 emissions instead of methane, we decided to delete the description of greenhouse gases and focused on the damming effect on CO2 emissions.

Comment 3: Thirdly, I think the authors need to better integrate the diel sampling component of their study into the way that the other results are analyzed. The authors don't mention the temporal sampling scheme employed during their 16 sampling campaigns-were sites always sampled in the same order? Over what range of times? Are we confident that variation in fluxes measured is more a function of spatial variation than temporal variation?

Response: Thank you for the kind reminding. We will add the information about the sampling timing. All the sixteen sampling campaigns were implemented in the daytime and basically followed the same order. Each campaign usually last for two days. In the morning of the first day (usually 9am), the sampling will be started from Point R1 to P2. Sampling in each point costed around 40 minutes so the sampling for the four points (R1, P1, L and P2) could be completed before 4pm. In the second day, the sampling starts at R2 (the time varied from 10am to 11am), following by Point D (1pm). In the third day, the sampling at Point P3 and P4 requires the boat so normally the campaigns were conducted in the afternoon around 3-5pm. We do not think the diel variation would overshadow the effect of seasonal variation as the sampling timing was similar in all campaigns.

We believed that the variation in fluxes is a function of both spatial and seasonal variation. There was significant variation in fluxes between riverine sites and the pelagic sites, but the variation is only significant in the dry season. In the wet season there is no significant spatial variation in fluxes between riverine sites and reservoirs sites (maybe the littoral area need to be isolated). It means that the extremely high emission rates only occurred in the riverine sites in the dry season. Since the sampling campaign spanned two to three days, different sampling dates might also lead to variation in fluxes. However, it should be noticed that average water retention time of the studied reservoir is 1.4 days. This type of daily-operated reservoir usually experienced repetitive fluctuation of water level in daily cycle, according to the operation guide of Chinese reservoirs. In the dry season, in particular, everyday the water level will be drained down to the lowest level (this is consistent with what we observe in the studied reservoir though we do not have enough data of water level to support it). In the same timing each day, the hydrological condition did not vary too much. Actually, the diel variation in water level might cause much more variation in fluxes than that sampling in different days.

Yet we cannot deny the effect of diel variation of hydrological condition on the variation of flux and that is why the diel sampling was conducted. However, our diel sampling did not capture the variation of CO2 emissions for a whole daily cycle. As shown in Fig. 5, the diel sampling did not cover the CO2 effluxes from 9am to 12noon, during which the sampling was conducted in the riverine sites. Due to the incomplete diel sampling, we had to average the daily flux and nocturnal flux respectively and integrated them into other results. We cannot calibrate the flux for all the sampling points according to the timing since sampling in riverine area were actually conducted in the period that we did not capture in the diel sampling. For the same reason we could not conclude that the flux was not independent of the sampling timing.

Given the significant spatial variation and temporal variation in fluxes, we examined and separate their effects with correlation analysis between flux and some controlling factors like light and heat (which was represented by water temperature). Firstly, we are confident that the light availability did not affect the spatial variation of fluxes since all the sampling were conducted when sunlight was sufficient. Secondly, we did not find the significant relation between water temperature and flux ($p > 0.10$) as the water temperature varied very little in the diel sampling. Instead, the relation between pCO2 and flux was significant (correlation coefficient = 0.665, $p < 0.001$). Thus, the diel variation in flux was attributed to the variation of pCO2. Assuming the measured fluxes in different points were caused by temporal variation, higher efflux must be resulted from higher pCO2. However, no significant relation between pCO2 and flux was found in the whole dataset and in all the grouping zones ($p > 0.10$). Yet the relation is significant in Point P4, though the correlation coefficient is negative. Therefore, we believe that some physical factors other than pCO2 caused the variation in different sampling points.

The timing and order of sampling campaigns will be added into the Methodology part and the correlation coefficient will be added into the Supporting Information for further clarification.

Comment 4: Fourth, while I think that hydrology may be a dominant control on reservoir CO2 emissions in this reservoir (e.g. it seems a completely valid and plausible hypothesis), I don't think the authors present enough evidence in support of this mechanism to present it as a result (e.g. in the abstract of the paper). Reservoir hydrology co-varies with other seasonal variation in temperature and the authors present no systematic approach for differentiating other possible controls.

Response: As our reply in Comment #3, we believed that there was supposed to be some factors other than temperature controlling the seasonal variation in flux at river inlet owing to the insignificant relation between water temperature and flux. However, the gradient in water temperature between inflow and receiving waterbody was significantly related to the flux in all the river inlet(p<0.001). Therefore, we speculated that the flux might be rely on the various mixing mode.

According to Summerfield (1991), the mode of mixing between sediment-laden river water and receiving waterbody was dependent on the relative water density. The hyperpycnal flow occurs when the incoming water was colder, denser and contained more suspended sediment loads than the receiving waterbody. On the contrary, hypopycnal flow occurs while the inflow was more warmer and clearer than the stagnant water in the receiving body. Here in Fig. 1&2 the flux was negatively related to the SPS concentration gradient at the riverine inlets. They showed that the high fluxes occurred when the inflow was warmer, less turbid and lighter than the receiving waterbody in the reservoir. As in dry season the inflow was warmer, less turbid and less dense than the reservoir water, we consider that the inflow became an overflow on surface and the higher pCO2 can enhance the emission rate. The situation was opposite in the wet season when the heavy turbid flow plunged into the reservoir bottom and short water retention time allowed little time for mineralization of organic carbon. The hypothesis was also supported by the negative relation between water discharge and CO2 flux (Fig. 3&4).

[Figure]

Fig. 1 The scatter plots showing the relation between CO2 flux and the difference in SPS concentration between riverine sites and reservoir surface at Point R1 (left panel, ln Flux = -0.331 $\Delta$SPS + 2.748, $R^2$ = 0.07, p<0.01) and at Point R2 (right panel, ln Flux = -0.332 $\Delta$SPS + 2.684, $R^2$ = 0.768, p<0.01).

[Figure]

Fig. 2 The scatter plots showing the relation between water discharge and CO2 flux at the Point R1 (left panel, p<0.01) and at the Point D (right panel, p<0.01) during the sampling period.

But considering the mixing mode and hydrological condition could be covaried with the water temperature, we will add scatter plots showing that the flux in river inlets were significantly (p<0.001) related to the gradient in suspended sediment concentration between the incoming water (represented by R1 & R2) and receiving waterbody (represented by P1 & P4) as well as the relationship between flux and water discharge (p<0.001) as evidence.

See the graphs here and Fig. 7 in the manuscript.

Comment 5: Finally, it is difficult to interpret the zonation grouping-the authors should consider incorporating a statistical assessment of significant differences between sites. For example, were the riverine samples from both sites more similar to each other than to other sites? Or was one riverine site emitting CO2 at much higher rates than the other? Reservoir inlets are often hot spots for biogeochemical activity-are we sure that these riverine sites are fully riverine and that their hydrology isn't influenced by the dam?

In addition to these scientific concerns, the manuscript needs to be edited for proper English. There are grammatical issues and vaguely written statements that could benefit from a third-party editor.

Response: We grouped the sampling points according to their surface velocity. The flow velocity in the surface water was 0.2m/s and 0.7m/s at the Point R1 and R2 respectively, according to the data measured in the preliminary fieldwork in this research in 2015). The Point D at the downstream of the dam also maintained a flow velocity but the flow was largely regulated by the dam. All the other points (including P1~P4 and L) was located within the backwater area and no flow velocity can be detected at the water surface at these locations. The fluxes from riverine sites were significantly different from the other sites (see Page 6, Line 17) but fluxes from R1 and R2 did not show significant difference (p>0.10). Therefore, we are confident that the fluxes at the riverine sites were significantly higher than the sites in reservoirs and at the downstream of the dam.

Generally, we considered that the water in the backwater area is stagnant in a reservoir and no flow velocity can be detected on its surface, even though subsurface flow could be maintained as the water was discharged to the downstream (as we put it in Page 3, Line 24). The boundary of backwater area is the boundary of a reservoir and the upper bound of the influenced area where the backwater pushed by the dam can influence the hydrological condition. But this boundary could be varied due to various water level. In this case, we

selected the points at the upstream of the boundary as the river inlet and minimize the effect from the Gongguoqiao Dam and guarantee the water was still flowing on surface. But unfortunately, we cannot really exclude the effect of the dam under construction at the upstream of the sampling points. We believed that the water discharge was not affected by the Miaowei Dam but the dam under construction might change the deposition processes of sediments.

We've tried to highlight the significant differences in the article. We will add the surface flow velocity to the introduction of riverine sampling points to validate the zonation grouping.

Page 1 Line 12: change "cycle" to "cycling"

Changed as suggested.

Page 1 Line 14: did the authors use a statistical approach to see if reservoir emissions were significantly different by season?

Yes. We used the Non-Parametric Tests (Independent Samples) to test the difference. The emission rates showed significant difference between the dry season and the wet season ($p < 0.001$).

Page 1 Line 17: I don't think the analysis presented here conclusively linked $CO_2$ emissions to physical mixing.

Even though the positive relation between water temperature gradient and $CO_2$ emission rate could hardly suggested the influence of different mixing mode, the relation between $CO_2$ emission rate and sediment gradient between river inlet and receiving waterbody and different seasonal variation trend of $CO_2$ emission rates and water discharge can link the emission to the mixing mode. Neither $pCO_2$ or water temperature cannot explain the seasonal variation of flux for riverine points. Thus, there was supposed to be physical processes affecting the emissions.

Page 2 Lines 3-5: Carbon dioxide is generally thought of as the largest contributor to total carbon emissions, but methane is generally the largest contributor to total greenhouse gas emissions on a $CO_2$ equivalent basis. I think the authors should be careful to make this distinction clear.

The sentence was revised as "Since carbon dioxide takes up largest portion in total carbon emission from inland waters". Because we are not going to present the data of methane flux, we will only emphasize the amount of the carbon dioxide in carbon emission while not consider the effect of methane.

Page 2 Line 18: By "biogeochemical processes of phytoplankton" do you just mean photosynthetic uptake?

Yes. As the word here is too vague, we changed the word into "photosynthetic uptake" as suggested.

Page 2 Line 24: The way you have phrased this sentence makes it sound like all the studies you are citing were conducted in the Three Gorges Reservoir, but Pacheco et al. 2014 was in Brazil. Also, I don't see Tao 2017 listed in your references section.

We broke this sentence into two sentences as "For example, the Three Gorges Reservoir… undersaturation of $CO_2$ in surface water (Zhao et al., 2011, Guo et al., 2011, Ran et al., 2011). The undersaturation could turn the fluxes…budget (Guo et al., 2011, Pacheco et al., 2014, Tao, 2017). The citation of Tao (2017) was added into the reference section.

Page 2 Line 25: Do you mean watershed? Not waterbody?

The article here refers to the eutrophication in the waterbody (see Page 2 Line 22). The stagnant tributaries which was impacted by the backwater can suffer severe eutrophication as the nutrient input cannot diffuse and thus cause algae bloom.

Page 3 Line 5: Why is information about Xiaowan Reservoir relevant here? Also, perhaps this is a good place to mention the construction of Miaowei Dam (which is noted in your Figure 1). Was the dam completed after your sampling ended in Dec 2016? Was the system hydrology affected at all by the fact that a dam was being constructed upstream during your study?

Because the outflow of Gongguoqiao Reservoir feeds directly into the Xiaowan Reservoir, we need to exclude the effect of backwater of Xiaowan Reservoir on the hydrological condition at Point D. We will add some introduction to the three reservoirs (Miaowei, Gongguoqiao and Xiaowan) here and supplement the detailed sampling timing and how we define the riverine sites. It will be highlighted here the sampling ended before completion of the Miaowei Dam. The construction of the dam, possibly impact the deposition of sediments in the riverine site but did not regulate the flow.

Page 3 Line 9: Is this a hydropeaking (load following) reservoir? It might be nice to see water level data from the reservoir given the current discussion of water level fluctuation you have incorporated into your discussion. Page 3 Line 16 (and throughout): You use "mainstream" when I think you mean "mainstem".

According to the meaning of hydropeaking reservoir we searched online, we confirmed that the reservoir is a hydropeaking reservoir. We will supplement the water discharge data to represent the variation of water level. We will replace the "mainstream" with "mainstem" throughout the article.

Page 4 Line 2: Consider reformulating the equation to take out unit conversion factors (which seem a little distracting and un-necessary).

The equation was quoted from the reference (Page 3 Line 30) but we can take out the conversion factor.

Page 4 lines 26-28: The authors discuss dam hydrology as if they don't know what type of spill practices are employed in the reservoir. Isn't this information available? Also, the height of reservoir spill (epilimnion versus hypolimnion) could be mentioned in the study area section.

Since the water level frequently fluctuated in the reservoir, the height of reservoir spill might be variable. But we did know that the water passing the turbine was drawn from epilimnion as the hypolimnion water was too turbid that it will harm the turbines. The staff from the reservoir told us that in the rainy season the water passing the turbine was drawn from a layer around 4m deep under the water surface. This is consistent with our observation that the flow velocity of subsurface flow was highest at around 5m deep at a high water level. The minor difference in water temperature between Point P3 and D was consistent with this spilling practice.

This information of spilling practice will be added into the introduction part of study area and the sentences at Page 4 Line 26-28 will be changed accordingly.

Page 5 line 3: Why do the authors feel that the dataset is limited? Is there reason to think that sometimes the running waters from inflow are not more aerobic than the reservoir water?

Theoretically, the running waters from inflow are more aerobic than the reservoir water. But since a dam was under construction at the upstream and the DO data was unavailable since October owing to malfunction of the instrument, we are uncertain about that in the dry season.

Page 5 line 11: Change this sentence to something like "With the exception of one sample, the reservoir was consistently supersaturated with CO2, indicating its role as a CO2 source to the atmosphere"

We changed it into "Most of the water samples had pCO2 higher than the atmospheric values.".

Page 5 Lines 20-24: A plot that shows water level and point CO2 measurements over time might be helpful here—I got a little lost in this description of the results.

We will add the plot showing the relation between CO2 flux and water discharge here.

See Fig. 2 above.

Page 7 Line 1-2: Where do the authors show this analysis? Right now there is no mention of a statistical analysis of drivers and no corresponding table or figure.

We will put some plots about the nutrient concentration and CO2 fluxes into supporting information.

Page 7 Lines 4-21: So, given these results, are you confident that the CO2 efflux measurements you made are still predominantly representing spatial (rather than temporal) variation? Also, it sounds like physical differences (rather than biology) may be driving the differential emissions you see during the day versus at night? Would you agree?

The results showed that the higher emission rates were only found at the riverine inlet in the dry season (spring and winter). The effluxes showed large diel variation, and this might affect the spatial variation. As the reply in Comment #3, we followed the same order of sampling at each site and tried hard to keep the same sampling timing in each campaign. Except the incontinuous sampling and the diel sampling, no sampling was conducted at night. Sampling campaigns by boat was conducted at different timing at P3 and P4, but we did not find any significant difference.

Page 7 Line 24: How do you define a pristine river channel? Was R1 at all influenced by the construction of Miaowei Dam? How do you differentiate free-flowing river from reservoir inlet?

We here define the pristine river channel as no dams at the upstream impounded the water and regulate the flow. We cannot deny that the flows at R1 could be influenced by the Miaowei Dam. Flow velocity and deposition processes might be affected since the river channel had changed. But since the dam did not regulate the discharge, we believe that the seasonal variation of water discharge would remained the same but possibly with less sediments.

We consider that the river reach with surface velocity over zero (v>0m/s) as free running river while the reservoir inlet was supposed to be a profile close to the boundary where the surface flow velocity decreased to zero. As the boundary of backwater area varied frequently due to the fluctuation of water level, we have been trying to select a sampling close to the boundary where the river was free-flowing as the reservoir inlet.

Page 8 Lines 11-12: I don't think Figure 7 really shows this.

The evidence is not sufficient here. We will add the graphs shown in Comment #4 and some discussions explaining how different mixing modes lead to the variation in fluxes.

Page 9 Line 8: Not sure "constraint" is the right word.

It refers to that the emissions at the downstream was kept at a low level. Possibly we can change the word to "restricted".

Page 10 Line 17: Why "potential"?

The word could be redundant and we will delete it.

Page 10, Conclusion: No discussion about why emissions were so high from the river in the dry season. Was this pattern consistent in both river sites?

We have tried to explain different mixing modes leading to the different seasonal variation in the CO2 fluxes in Page 10 Line 22-25. Possibly the explanation was not clear enough. With the correlation analysis between CO2 effluxes and water discharge as well as SPS concentration, we can explain the modes and how they influence the CO2 fluxes much clearer.

Page 10, Line 31: What pattern are you referring to?

This pattern refers to the higher emission rates in the dry season. We will explain that in the sentence.

Figure 6: Continuous versus incontinuous diel sampling was not explained in the methods.

A continuous diel sampling on CO2 efflux was conducted before the last sampling campaign. Besides, incontinuous sampling for the diurnal variation in fluxes were also conducted in the riverine sites during the first sampling campaign in January. The information will be added into the Sampling section.

GUO, J., JIANG, T., LI, Z., CHEN, Y. & SUN, Z. 2011. Analysis on partial pressure of CO2 and influencing factors during spring phytoplankton bloom in the backwater area of Xiaojiang River in Three Gorges Reservoir. *Advances in Water Science,* 22**,** 829-838.

PACHECO, F. S., ROLAND, F. & DOWNING, J. A. 2014. Eutrophication reverses whole-lake carbon budgets. *INLAND WATERS,* 4**,** 41-48.

RAN, J., LIN, C., GUO, J., CHEN, Y. & JIANG, T. 2011. Spatial and temporal variation of carbon dioxide partial pressure over the Xiaojiang River backwater area of the Three Gorges Reservoir. *Resource and Environment in the Yangtze Basin,* 20**,** 976-982.

SUMMERFIELD, M. A. 1991. *Global geomorphology: an introduction to the study of landforms,* Harlow, Essex, England;New York;, Longman Scientific & Technical.

TAO, F. 2017. Air–water CO2 flux in an algae bloom year for Lake Hongfeng, Southwest China: implications for the carbon cycle of global inland waters. *Acta Geochimica,* 36**,** 658-666.

ZHAO, Y., ZENG, Y., WU, B. F., WANG, Q., YUAN, C. & XU, Z. 2011. Observation on greenhouse gas emissions from Xiangxi River in Three Gorges Region. *Advances in Water Science,* 22**,** 546-553.

**#2**

This study reports CO2 emissions from a hydroelectric reservoir located in the upper
Mekong river basin
The writing style makes difficult the review of this article.
The site description is incomplete:
1 What was the land cover before flooding?

Response: The right bank of the reservoir was relatively flat, and some small villages sparsely was built at
this side. Before flooding, the right bank of reservoir was mostly farmland, uncultivated grassland and the
residential lands of villages. The left bank was steep, and landslides frequently occurred in rainy season.
Thus, the left bank was either grassland or barren land. Before the reservoir filling, the cover of
vegetation accounted 10~20% of the reservoir valley.
2 What are the water discharge in and out of the reservoir according to seasons? –What is the average
water residence time?

Response: The water discharge at the river inlet R1 and R2 in each season can be found in the Fig. 1&2 in
this letter. The average water residence time was 0.01 year (Shi et al., 2017)  or 1.4 days (See Page 3 Line
9).The inflow of the mainstem and tributary could be found in the figure below (the left panel is the water
discharge in the mainstem while the right panel is the inflow from tributary). Noticed that the inflow of
the tributary was extrapolated with the instantaneous water discharge during the sampling at Point R2. We
have highlighted the daily-operated nature (active storage/mean flow discharge <0.08) and short water
retention time in the introduction of the reservoir. The flow can be regulated in a daily cycle while hardly
be changed at seasonal scale.  Currently we only have the instant water discharge at the outflow during
the sampling campaign.

3 Importantly, the seasons must be described precisely and the same nomenclature must be used
throughout the article instead of using sometimes summer, winter, spring. . . and at other places warm
season, rainy season and even some combination like warm dry seasons. . . The reader is lost. . .

*Response: The wet (rainy) season spanned from May to October every year while November to April in
next year was considered as the dry season (see Page 3 Line 12). When under the control of South Asia
monsoon climate, the rainy season is usually warm while the dry season is cold as it usually covers winter
and spring. Considering the samples from winter were fewer than that from other seasons (as we have to
finish the campaign before the filling of the Miaowei Dam at the upstream) and this could lead to bias in
statistics, we combined the dataset according to the distinctive hydrological condition and rainfalls in the
wet season and the dry season. Yet in autumn when the wet season came to an end, some emission rates
exhibited some different characteristics from other seasons. Thus we presented the data of pCO2 and
emission rates in four seasons and separated some extreme high pCO2 and emission rates from other
seasons. Yet we will add the detailed classification and the characteristics of the monsoon climate will be
to the description of study area.*
4 Meteorological information like temperature and rainfall range are required

Response: We will put the basic information into the introduction of the reservoir, including monsoon,
precipitation, air temperature and land covers.
5 the map (figure 1) requires a scale, an orientation and information about direction of
the water flow would be welcome.

Response: We can add the scale but the scale and orientation can be read from the coordinates marked at the outline of the map. The direction of flow was from North to South at both mainstem and tributary, which is consistent with the flow direction of Mekong River flows from Tibetan Plateau to South China Sea. Later we will add the catchment map of the Mekong River and highlighted the position of the reservoir in Fig. 1.

6 is the reservoir thermally stratified? Well mixed? Monomictic?... Such information is required to be able to understand the seasonal dynamic of a lake or a reservoir

Response: Insofar there was no reports on the stratification situation for the reservoir. But we measured the vertical profile of water temperature at Point P3 in some sampling campaigns. It was found the water was well mixed from May to August while in the rest of the year, stratification developed in the pelagic area where water depth is over 5m. The water temperature dropped drastically at 5m below the surface and stabilized deeper onward. The difference in water temperature between surface water and sediment surface was around 2 °C. We can only add some basic statistics data on vertical profiles of water temperature from this research as supplements as we are still examining the quality of vertical datasets.

The sampling strategy requires clarification

7 Can we call the station L as a littoral station since it seems to be an artificial island which has developed after sedimentation in the reservoir? In some part of the manuscript it is also called the drawdown area. . . Again, the reader is lost by the inconsistence of the vocabulary.

Response: The Point L is a wetland in a reservoir bay formed after impounding due to sedimentation. We will unify the name for consistency.

8 P3-L19 stations P1 to P4 are considered all together whereas a few lines below, only P2-P4 are considered as pelagic stations. What type of station P1 is representative for?

Response: Possibly there was some misunderstanding due to the order of introduction. Pelagic points include P1 and P2-P4. The Point P1 was located within the reservoir as no surface velocity was detected here. As the point was permanently flooded, we considered the point as a pelagic point. Of course the Point P2-P4 were also classified as pelagic points as they were also permanently and had no surface velocity.

9 Not clear in the sampling strategy and site description but the sampling occurred during the year 2016 (P2L23) while the dam upstream of the study site was completed by December 2016 (Figure 1). Therefore, all the sampling might have been done during the construction which means that the river was heavily disturbed. The construction might have biased the conclusion on the fact that the "pristine river" (as the authors call it) emits more than the reservoir itself.

Response: Yes. The sampling campaigns were completed before the filling of the reservoir. We cannot deny that the construction at the upstream might have disturbed the river. However, as the natural flow was not regulated by the artificial dam, we assume that the river was free running and its hydraulic regime remained the same when the reservoir has not been filled. But as the dam slowed down the flow velocity, the turbulence resulted from higher surface flow velocity can be reduced and thus emission rates could be decreased. Hence possibly the emission rates at the pristine river could also be underestimated and the conclusion could be right but conservative, though the bias might exist. As the grouping in the manuscript might be confusing, we will try to clarify the standards in the selection of sampling points.

The methodology is minimalist and substantial information is missing to be able to evaluate the quality of the dataset:

10 How many samples were gathered in total? By campaigns? Was the sampling organized by seasons?

Response: We are sorry that we did not make it clear in the introduction of sampling scheme. The formal sampling campaign started from April to December, 2016. Totally sixteen sampling campaigns were conducted on the eight sampling points, with a frequency of twice a month. During the formal sampling campaign, 127 samples were collected as we failed to gather the water samples from the Littoral zone in one campaign in October as the area was totally drained at the low water level. Another two preliminary campaigns were conducted in January and March respectively, in which only the riverine points were sampled. We will add these information to the sampling introduction.

11 -P3-L25-30: what are the precision, range and accuracy of the gas analyser? What gas flow was used? Did the author used desiccant? Is there a humidity correction is the analyser? What is the volume of the chamber? How were measured the fluxes in the river? At fixed station or drifting with the flow? What was the rejection/acceptance procedure for the measured fluxes?

Response: The portable S157 CO2 Analyzer produced by Queen's University Biological Instrument & Technology (Qubit, Canada) was used to measure the CO2 concentration. The S157 CO2 Analyzer is a single channel non-dispersive infrared CO2 analyzer that measures CO2 in 0 to 2000 ppm range with 1 ppm resolution. The built-in pump in the analyzer directly draws the air for analyzer and the desiccant was installed within the intake tube. More information on the analyzer is available at the following website:

https://qubitbiology.com/s157-co2-analyzer-0-2000ppm/

The volume of the chamber is $2400cm^3$ as its height, width and length can be found in Page 3 Line 25. When measuring the fluxes in the river, the chamber was floating and fixed to the piles marking water levels. Generally, we waited for the stabilization of the analyzer to a range of 400~500ppm (atmospheric pCO2) and kept monitoring the variation of pCO2 in the chamber for 15~20 minutes via the laptop. The curve was accepted and used for calculation of fluxes once R square reached 0.90. Great fluctuation of concentration was rejected and the measurement would be restarted again. As the properties of the analyzer could be easily found in the company's website, we do not think we need to list them in detail. Details of rejection and acceptance procedure can be found in Tremblay et al. (2005) as we cited.

12 What are the precision and accuracy for Temp, O2, pH, conductivity measurements? This is critical for pH since pCO2 was calculated by pH/Alka method. Details on pH measurements are required

Response: The precision of water meter for Temp, O2, pH and Cond are 0.1℃, 0.01mg/L, 0.01 and 0.01μS/cm respectively. The meter was calibrated according to the manual before each campaign begins and the properties were measured three times for an average. The probe for pH was calibrated with three standard solution (pH = 4, 7, 10 respectively) before sampling and the pH would be tested with the neutral solution to examine the accuracy. The pH was generally higher than 8.0 in the Lancang River. But sometimes lower value (<7.0) was also found, we clear the probe and retested the pH. If the value showed consistent results in four times of measurements, the value will be accepted. Since we followed the standards of calibration and measurements of these water properties, we believe the measurement of pH should be accurate. Information and manual can be easily attained online as we presented the type and company of the meter.

13 Precision and accuracy and detection limit are required for Alkalinity.

Response: Water samples were titrated with the HCl solution (see Page 4 Line 9-10) to the point that methyl orange turned orange. The concentration of HCl solution was titrated with NaOH solution each time the acid was prepared. The average concentration of the HCl solution was around 0.024 mmol/L. The precision of 2mL burette used in the titration of water samples is 0.01mL. Therefore, the precision of

alkalinity was supposed to be 0.024mmol/L and any alkalinity lower than the value could not be detected. The titration is a popular way to measure the total alkalinity so we do not think we need to explain the solution of acids and discuss the accuracy with too much details in Methodology.

14 pCO2 using pH, Alkalinity and the CO2SYS program. This validity of the methodology was discussed recently by Abril et al. (2015) and (Golub et al., 2017) for inland waters

Response: We also noticed that the CO2SYS program might overestimate the system. Since the pH has largest weight in the program, even a slight variation in pH could lead to drastic fluctuation in pCO2. However, we do not think that the selection of methods for pCO2 calculation influenced or contradicted our conclusion that the high emission rates were caused by physical factors.

Firstly, the measurements of CO2 emission rates did not rely on the calculated pCO2. The parameters that used for the calculation (alkalinity, temperature and pH) was totally independent from the measurements of CO2 fluxes. As the article emphasizes the importance of hydrological condition and mixing mode in regulating the CO2 emissions at the river inlets and reservoir surface, rather than the pCO2 in surface water. Even though we tried to calculate the outgassing rates with pCO2 and gas transfer rate (Thin-Boundary Layer Model), we finally decided not to include the datasets into the article but simply present an average as a comparison as we noticed that the dataset could be bias.

Secondly, as the referee cited from Abril et al., (2015), the calculated pCO2 could be largely overestimated in the acidic and organic-rich waters. But in the GGQ Reservoir, even the highest DOC concentration was no more than 2.992ppm (Point P1). Besides, the pH of the Lancang River was always higher than 8.3 (See Table 1), suggesting the environment in the reservoir was alkaline. In such alkaline and organic-poor system, fluctuation of pH could hardly make significant variation in pCO2. Sometimes we also recorded an abnormal increase at some sampling sites as the drifting deadwoods tends to release organic acids during decomposition (as we highlighted in Page 9 Line 14). The abnormal points were separated from the dataset for discussion as they can interfere the results and not quite related to the conclusion.

Thirdly, given the random error and systematic errors in the calculated pCO2, the variation of pCO2 might remained the same after excluding the abnormal value as it was used to explain the spatial and temporal variation of the flux. When the correlation between pCO2 and CO2 fluxes were analyzed, the systematic error could hardly cause great bias as the procedure determining the pCO2 was consistent and the aquatic environment did not exhibit large heterogeneity in alkalinity (maybe not applicable to the littoral zone so we separate the point from pelagic area), which might cause the bias in pCO2 calculation according to Golub et al. (2017). Finally, although the head-space equilibrium method could be a better way to measure the pCO2, most of the existing studies on pCO2 in Chinese reservoirs (and sometimes rivers also) used the calculated pCO2 and the inconsistent method possibly impede the comparison to other reservoirs in China and incorporation into the existing database.

15 For chlorophyll: How long after sampling the water was filtered? Were the filters kept in the freezer? What was the precision, accuracy and limit of detection for Chlorophyll, DOC, TOC, TN and TP?

Response: The infiltration for chlorophyll started four hours after the sampling campaign finished. The filters were kept in refrigerators. In this study, the precision of the chlorophyll concentration was 0.01mg/L, even though the instrument could detect lower concentration down to 0.1µg/L. Calibration was conducted before the analysis by technician according to manual and the details can be found in the following link:

http://www.walz.com/downloads/manuals/phyto-pam/PhytoPamII_2.pdf

The precision of DOC concentration was 0.001ppm. Standard samples with a concentration of 1, 2, 4, 5, 7, 10ppm would be tested for a standard curve before the analysis on water samples. The curve was accepted when the R square of regression reached 0.95. Before analysis blank samples (pure water) would be tested first for subtraction. A standard sample was inserted into the sequence with every 10 samples to monitor the operation of instrument. The attained results will be calibrated with the standard samples after the subtraction of blank values.

The procedure we followed when measuring TN and TP was the unified standards for the surface water on earth in China. The analysis of TN and TP was similar to that of DOC with the same subtraction of blank samples and calibration with standard samples. The standard curve was only accepted when the R of linear regression reached 0.999. The precision for TN was 0.05mg/L and the limit of detection was 0.20mg/L. The precision and detection limit for TP was 0.01mg/L.

The methods, precision, and detection limit of TN and TP can be easily found online. As it was long and easily accessible, we are not going to add it into the methods.

We did not publish any TOC data in this article. Please check it again.

16 statistics used for the seasonal and spatial variations were not described the thin boundary method which was used according to P6L10 was not described

Response: The methods was cited from Goldenfum and Association (2010) and we assumed an average atmospheric pCO2 of 406μatm. Like the CO2 efflux, significant difference in the outgassing rates was found between riverine sites and reservoir sites (p<0.01) but the spatial variation was insignificant within the reservoir (p>0.10). No other significant spatial or temporal variation was found in the outgassing rates as it showed quite homogeneous value throughout the year and the reservoir sites. The results and statistics of outgassing rates calculated with the Thin-Boundary Method were deleted because its seasonal and temporal variation was quite similar to that of pCO2. The pCO2 weighted too much in the calculated flux and dominates its variation. Since we have noticed that the calculated pCO2 could be bias, we decided not to discuss the seasonal variation of calculated outgassing rates furtherly but only present an average of these results as a comparison. The calculated rates, however, can be presented in the supplements in case some readers are really interested in it.

According to the fact that the sampling strategy and the validity of the pCO2 dataset is doubtful and the quality of dataset cannot be evaluated in absence of information, it is impossible to go further with the review of this manuscript.

Response: We appreciate the referee's reviewing and questioning on the methods applied in the research. We supplemented some information to make the method clearer and further verify the dataset we collected.

Abril G, Bouillon S, Darchambeau F et al. (2015) Technical Note: Large overestimation of pCO2 calculated from pH and alkalinity in acidic, organic-rich freshwaters. Biogeosciences, 12, 67-78.
Golub M, Desai AR, Mckinley GA, Remucal CK, Stanley EH (2017) Large Uncertainty in Estimating pCO2 From Carbonate Equilibria in Lakes. Journal of Geophysical Re

GOLDENFUM, J. A. & ASSOCIATION, I. H. 2010. *GHG Measurement Guidelines for Freshwater Reservoirs: Derived From: The UNESCO/IHA Greenhouse Gas Emissions from Freshwater Reservoirs Research Project*, International Hydropower Association (IHA).

SHI, W., CHEN, Q., YI, Q., YU, J., JI, Y., HU, L. & CHEN, Y. 2017. Carbon Emission from Cascade Reservoirs: Spatial Heterogeneity and Mechanisms. *Environmental science & technology,* 51**,** 12175-12181.

TREMBLAY, A., VARFALVY, L., ROEHM, C. & GARNEU, M. 2005. *Greenhouse gas emissions-fluxes and processes*, Springer.

**List of Relevant Change**

**Introduction:**

1.  According to the comments of associate editor, the significance of methane in greenhouse gas emissions from inland waters is admitted and we changed the way to emphasize the importance of $CO_2$ emissions from reservoirs (Page 2, Line 11-13).
2.  Since there are already a number of studies on the dynamics of carbon dioxide emissions from the reservoirs in China, some special dynamics underlying variation of emissions are reviewed (Page 2 Line 17 to 22) with the example of Three Gorges Reservoir (Page 2 Line 26 to 29). According to the comments from Referee #1, the monsoon climate in the upper Mekong River Basin is briefly introduced and the how damming will impact the carbon dioxide evasion under such monsoon climate is emphasized in Page 3 Line 1 to 11. The study objectivity is also revised accordingly (Page 3 Line 13 to 16).

**Methodology:**

1.  To give a more detailed description of the studied reservoir, we add the catchment area, annual water discharge, seasonal variation of water discharge and sediment loads, average precipitation, average air temperature and average water retention time at Page 3 Line 20 to 28. The monthly water discharge of inflow of the reservoir is illustrated in Fig. 2, and the time span of rainy season is added (Page 3 Line 24).
2.  Besides, as sampling map might not make the location of the reservoir clear, we add one map (shown at the lower right corner of Fig. 1) highlighting the studied reservoir in the Mekong River Basin.
3.  We added the point that the outflow of the reservoir feed in the Xiaowan Reservoir at the downstream (Page 3 Line 29) and only epilimnion water would be used to generate hydroelectricity (Page 4 Line 1).
4.  Stratification condition of the reservoir is supplemented in Page 4 Line 3 to 4.
5.  Details of sampling point grouping is given in Page 4 Line 8. The average flow velocities at the riverine inlets is listed to explain why we considered the riverine points as pristine river channels. The condition of littoral zone is described in Page 4 Line 16 for a better understanding and the term "drawdown area" is replaced to avoid inconsistency.
6.  The timing and sampling schemes are added into the Methodology (Page 4 Line 19 to 27), including the preliminary studies and formal sampling campaign. We highlighted here that the all the campaigns were accomplished before impounding of the Miaowei Reservoir and all the campaigns were conducted in daytime. Details of sampling for diel variations, including continuous sampling and discontinuous sampling, is introduced in Page 4 Line 24.
7.  Details of measurements of CO2 emission rates is added into Page 4 Line 32, including the resolution and stabilization of the analyzer and the floating condition of the chamber.
8.  The conversion factor in the equation calculating the efflux is deleted (Page 5 Line 5) and the acceptance criteria of slope is added (Page 5 Line 7).
9.  The resolution of water temperature, pH, conductivity and dissolved oxygen measurements were added (Page 5 Line 9).

10. The resolution of chlorophyll and DOC concentration measurements were added (Page 5 Line 22).

**Results:**

1. Some analysis of the monthly variation of the air temperature (as shown in Fig. S2) and the water discharge (as shown in Fig. 2) is added (Page 6 Line 3).
2. Some expression showing the difference in water temperature and dissolved oxygen between upstream and downstream is revised according to the comments of Referee #1 (Page 6 Line 10 and Line 16)
3. High pH value is highlighted for low pCO2 (Page 7 Line 3) and its relation to pCO2 is emphasized, such as in Page 7 Line 10).
4. The comparison between measured efflux and calculated outgassing rates is deleted (Page 7 Line 26).
5. Some correlation analysis on the diurnal variation of pCO2 is added (Page 8 Line 27 to 30) as evidence to support the integration of diel sampling component when the results were extrapolated. Statistical of pCO2 is also supplemented (Page 9 Line 6).

**Discussion:**

1. Correlation analysis between water temperature and pCO2 and the implication is added (Page 10 Line 1 to 4), and statistical analysis of efflux were added to highlight the high emission rates of riverine inlets in the dry season (Page 10 Line 6 to 9).
2. Scatter plots showing the different mixing modes (Fig. 10) and the correlation between water discharge and efflux (Fig. 9) were added as evidence to support the conclusion that the seasonal variation of efflux at riverine inlets were caused by different mixing modes (Page 10 Line 14 to 18).
3. A scatter plot showing the positive relation between TN concentration and efflux at the littoral zone is added into the supplemental information (Fig. S2) to explain the relation between pCO2 and eutrophication (Page 11 Line 10 to 12).
4. The annual emission rate of carbon dioxide is revised as we found some error in the original calculation process (Page 11 Line 17 and Line 20).

**Conclusion and Abstract:**

Since the contents in the results and discussion have been revised, the structure and organization of conclusion and abstract have been revised accordingly, but the conclusion remained the same.

**Figures and Table:**

1. A map indicating the location of the studied reservoir within the Mekong River Basin is added into Fig. 1 for clarification.
2. Fig. 2 is added to display the variation of water discharge of inflows at the mainstem and tributary.

3. Fig. 9 is added to display the negative relation between water discharge and efflux at the riverine inlet and outlet.
4. The scatter plots showing the relation between efflux and gradient in suspended sediment concentration between riverine inlets and reservoir surface were added to the Fig. 9 to reflect the impact of mixing modes on efflux.
5. Table 1 is adjusted to fit the window.
6. Fig. S1 showing monthly variation of precipitation and air temperature is added to the supplemental information. Fig. S2 showing the correlation between total nitrogen concentration and efflux at the littoral area is also added to the supplemental information.
7. The matrix of correlation coefficients is listed in Table S1 for reference.

**Other revision**

Some minor revisions on language are not listed in detail. The word usages highlighted in the reviews are changed as suggested and some vague sentences are revised as well.

**Physical-controlled CO₂ effluxes from reservoir surface in the upper Mekong River Basin: a case study in the Gongguoqiao Reservoir**

Lin Lin[1], Xixi Lu[1, 2, *], Shaoda Liu[3], Shie-Yui Liong[4] and Kaidao Fu[5, *]

[1]Department of geography, National University of Singapore, 117570, Singapore

[2]Inner Mongolia Key Lab of River and Lake Ecology, School of Ecology and Environment, Inner Mongolia University, Hohhot, Inner Mongolia, 010021, China

[3]Yale School of Forestry & Environmental Studies 195 Prospect Street New Haven, CT 06511. USA

[4]Tropical Marine Science Institute (TMSI), National University of Singapore, 117570, Singapore

[4]Asian International River Center, Yunnan University, Chenggong University City, Chenggong, Kunming, Yunnan, 650500, China

*Correspondence to*: Kaidao Fu(kdfu@ynu.edu.cn)

**Abstract.** Impounding greatly alters the carbon transportation in rivers. To quantify this effect, we measured $CO_2$ effluxes from a mountainous valley-type reservoir in the upper Mekong River (Lancang River in China) and compared them with those from the river channel. Evasion rates from the reservoir surface were $408\pm337$mg m$^{-2}$ d$^{-1}$ and $308\pm261$mg m$^{-2}$ d$^{-1}$ in the dry season and the rainy season respectively, much lower than those from riverine channel of $2168\pm2567$mg m$^{-2}$ d$^{-1}$ and $364\pm195$mg m$^{-2}$ d$^{-1}$at the mainstem and the tributary respectively. Low effluxes in pelagic area resulted from few allochthonous organic carbon (OC) inputs and photosynthetic uptake of $CO_2$. The negative relation between efflux and water temperature suggests that $CO_2$ emissions at the pelagic area were partly offset by photosynthesis in the warmer rainy season. The emissions from the reservoir outlet and the littoral area, which were usually considered as hotspots of $CO_2$ emissions, contributed little to the total emission because of epilimnion water spilling and small area of littoral zones. Yet the higher effluxes were recorded at the river inlets in the dry season when the inflow and outflow were small because of different mixing modes occurring in the two seasons. When the river joined the receiving waterbody in the dry season, the warmer, clear and lighter inflow became an overflow and substantial $CO_2$were released to the atmosphere as the overflow contacted the atmosphere directly. Extended water retention time due to water storage might also help mineralization of OC. In the wet season, however, the colder, turbid and heavier inflow plunged into the reservoir and was discharged to the downstream with carbon for hydroelectricity, leaving insufficient time for decomposition of OC. Besides, diurnal efflux variability indicated that the effluxes were significantly higher in the night than in the daytime, which increased the annual emission rate by a half.

**1 Introduction**

Supersaturation of $CO_2$ in the inland waters (Cole et al., 1994) results in substantial carbon outgassing to the atmosphere annually (Battin et al., 2009; Cole et al., 2007; Raymond et al., 2013; Tranvik et al., 2009). Loss of carbon to the atmosphere from inland waters has been recognized as an important part of carbon cycling which faces great anthropogenic impacts (Maavara et al., 2017; Regnier et al., 2013). Damming rivers to build large reservoirs for water supply, irrigation, hydroelectricity and flood controls is one of the most drastic changes in inland waters (Lehner &

Döll, 2004; Varis et al., 2012; Yang & Lu, 2014). By flooding large area of forests, soils and different kinds of organic matter, reservoirs have been identified as a large potential carbon source to the atmosphere since last century and have caused a serious perturbation on the global carbon budget (Fearnside, 1997; Kelly et al., 1994; Rudd et al., 1993). Damming rivers not only enlarges the water surface, but also produces more greenhouse gases (GHGs), mainly carbon

5    dioxide and methane, than the natural waterbodies (Barros et al., 2011, Deemer et al., 2016, Mendonça et al., 2012a). Most of the carbon is released in the form of carbon dioxide, even though methane takes up the majority of the GHG emissions (calculated with $CO_2$ equivalents) due to its high global warming potential (GWP) (Deemer et al., 2016, Demarty and Bastien, 2011).

10   Efforts have been made to evaluate $CO_2$ emissions from reservoir surfaces (Raymond et al., 2013; Varis et al., 2012; Vincent et al., 2000) and the accumulated case studies indicate that $CO_2$ emission rates exhibit great seasonal variability and spatial heterogeneity (Barros et al., 2011; Deemer et al., 2016). Quantity and quality of DOC and water temperature are considered as the most important factors that control the $CO_2$ fluxes from reservoirs as young tropical reservoirs and those with substantial labile OC tend to have higher emission rates (Barros et al., 2011; Mendonça et

15   al., 2012a; Tadonleke et al., 2012). However, in China, the country with the most dams in the world (Yang et al., 2016), analysis on $pCO_2$ shows that most of the effluxes from reservoir surface were much lower than that from tropical and boreal reservoirs (Li & Zhang, 2014; Li et al, 2015; Liu et al., 2016b; Ran et al., 2017). Lower effluxes in the reservoir center (Gao et al., 2017; Mei et al., 2011; Liu et al., 2016b; Liu et al., 2017) imply that the $pCO_2$ in reservoir surface is subject to photosynthetic uptake of phytoplankton (Ran et al., 2017; Ran et al., 2018). The $pCO_2$

20   and effluxes from reservoirs are regulated by the balance between respiration and photosynthesis and quite sensitive to the monsoon climate due to the seasonal variation of water temperature and hydrological condition (Guo et al., 2011; Mei et al., 2011). For example, in the Three Gorges Reservoir, one of the largest reservoirs in China, $CO_2$ emissions from the littoral zone are subjected to the seasonal flooding (Chen et al., 2009; Yang et al., 2012) and the carbon uptake of algae in the stagnant tributaries resulted from heavy eutrophication, was heavily influenced by the

25   seasonal variation of hydrological condition (Jiang et al., 2012, Guo et al., 2011, Ran et al., 2011, Zhao et al., 2013)

Despite the spatial heterogeneity (Li & Zhang, 2014), the research reviewed above mostly focused on the reservoirs in the highly populated eastern plain where the waterbodies are suffering from heavy eutrophication (Li & Zhang, 2014; Mei et al., 2011). In the less populated southwestern China where two-thirds of the exploitable hydropower were found and many more reservoirs are being built, however, the dynamics underlying $CO_2$ emissions has been less

30   understood (Hu & Cheng, 2013). Rivers originate from the Tibetan Plateau and flow through the mountainous area of Southwestern China, receiving flows from melted glaciers and rainfalls brought by the South Asian monsoon. The precipitation in summer and autumn account for 50% and 27% of the annual rainfalls respectively, producing high waterflow in the warm rainy season. It was supposed that the $CO_2$ emissions of these rivers are more sensitive to the

35   monsoon climate which regulates rainfalls, nutrient availability, and water discharge. However, the river flows are also regulated by the dams. In particular, dams completed upon the upper basin of Mekong River (or the Lancang

River), one of the most important rivers in Southeast Asia, have largely affected the hydrological condition, sediment transportation and the $CO_2$ emissions (Lu and Siew, 2006; Lu et al., 2014).

In this study, the Gongguoqiao Reservoir (GGQ), the uppermost reservoir in the Lancang cascading reservoir, was selected as a site for the investigation of the seasonal variation of the dynamics of carbon effluxes in these reservoirs. This research aimed to measure the $CO_2$ evasion with static chamber method and analyze the spatial heterogeneity, seasonal variation and diurnal variation of the $CO_2$ efflux, in order to examine the mechanism that controls the $CO_2$ effluxes under the monsoon climate and the damming effect on carbon emissions. Considering there are seven completed dams on the upper Mekong Basin and another fourteen dams are either under construction or planned, clarifying the coupling effect of the climatic and damming effect on the $CO_2$ emissions can help understand the role of inland waters in the global carbon cycle.

**2 Methods**

**2.1 Study area**

The Gongguoqiao Reservoir (GGQ) is located in Gongguo Town (Fig. 1, 25º35'9.87"N, 99º20'5.55"E) in Dali Prefecture (Yunnan, China).With a catchment area of 97,200 km², around 32 billion m³ of water flow into the reservoir annually. The monthly water discharge of inflow to the GGQ Reservoir in 2016 is shown in Fig. 2. Point L (Jiuzhou) is considered the point dividing the upper and middle reach of the Lancang River (Fig. 1). The area is subject to a subtropical monsoon climate where over 80% of the annual rainfalls bring 78.6% of the annual water discharge and 95% of the annual sediments loads to the reservoir in the rainy season spanning from May to October (Fig.2,He and Tang, 2000).The annual precipitation is 804.90mm and the monthly air temperature ranged from 7.6 ℃ to 21.6 ℃, with an average of 17.8 ℃ (Fig. S1, Hu, 2010). There are several villages scattered along the riverside. Before the reservoir filling, the average vegetation covered25% of the steep slope but the vegetation keeps degrading due to intense agricultural activities (Hu, 2010, Xu et al., 2003). The reservoir was filled in Sep 2011 and had been the uppermost cascading reservoir in the upper Mekong River Basin until the end of 2016 when the Miaowei Reservoir was filled at its upstream. The outflow from GGQ feeds the Xiaowan Reservoir at the downstream. The backwater area stretches 44.3 km along the mainstem and 7 km along the tributary, the Bijiang River respectively. The width of the reservoir ranges from 110m to 120m in the dry season. The standard water level is 1307m, corresponding to a storage of 0.316 billion m³. The reservoir uses epilimnion water (around 4~5m deep) for hydropower production and generates 4.041 billion kW/h annually. The reservoir is a daily-operated reservoir due to its small operating capacity (49 million m³). Thus, the water level fluctuates frequently and the average water retention time is 1.4 days. Water column is well mixed in the deep pelagic area (depth>5m) from May to August while stratified in the rest of the year (unpublished data in this research).

**2.2 Study methods**

**2.2.1 Sampling**

[revised manuscript text omitted]

**3 Results**

**3.1 Spatial and temporal variation of environmental factors**

Seasonal variations of temperature and rainfall reflect the characteristics of monsoon climate (Fig.S1). In winter (from December to February), the air temperature was below 5 ℃ while the monthly average temperature was all over 25 ℃ in summer (from June to August). The peak discharges of inflows in mainstem and tributary were both recorded in July, which were $70.50*10^8 m^3$ and $4.02*10^8 m^3$. The inflows in summer accounted 47% and 65% of the annual discharge in mainstem and tributary respectively. The water in the inflow 
[revised manuscript text omitted]

**3.4 Diurnal variation of $CO_2$ effluxes**

In GGQ the effluxes showed significant difference between daytime and nighttime (p<0.01). The diurnal observation of effluxes in the littoral zone showed that the $CO_2$ efflux was two times higher at night (from 19:00 to 7:00: averagely $495 \pm 178$ mg m$^{-2}$ d$^{-1}$) than in the daytime (from 7:00 to 19:00: averagely $247 \pm 171$ mg m$^{-2}$ d$^{-1}$) (Fig. 6 & Fig. 7). The $CO_2$ efflux was two times higher at night (from 19:00 to 7:00: $495 \pm 178$ mg m$^{-2}$ d$^{-1}$ on average) than in the daytime (from 7:00 to 19:00: averagely $247 \pm 171$ mg m$^{-2}$ d$^{-1}$). The trend was verified by the discontinuous efflux measurements in which the nocturnal $CO_2$ flux ($1012.29 \pm 1016.84$ mg m$^{-2}$ d$^{-1}$) was higher than the daytime flux ($766.87 \pm 740.43$ mg m$^{-2}$ d$^{-1}$). The efflux was negatively related to air temperature, wind speed and pH, but positively related to conductivity, alkalinity and $pCO_2$ (N=40, p<0.01). Thus higher efflux at night was resulted from dominated respiration in the surface water when light was unavailable for photosynthesis, which was also commonly found in other reservoirs (Liu et al., 2016a; Peng et al., 2012; Schelker et al., 2016).

Fig. 6 shows that $pCO_2$ was higher with an average of 969 µatm at night, but lower with an average of 871 µatm in the daytime. However, there was drastic oscillation of efflux from 9pm to 11pm with a range spanning from 712 mg m$^{-2}$ d$^{-1}$ to 69 mg m$^{-2}$ d$^{-1}$. Before 8pm, the efflux was kept below 400 mg m$^{-2}$ d$^{-1}$ but rose to above 450 mg m$^{-2}$ d$^{-1}$ after 0:30 at midnight. Statistically there was no significant difference in $pCO_2$ between nighttime and daytime (p>0.50).

The diurnal variation in $pCO_2$ was also insignificant because the pH varied little within a daily circle (p>0.50). The pH was 8.21 on the average with a range of no more than 0.28. However, a slight decrease in pH was found at night, which led to an increase of $pCO_2$ and efflux. The water temperature increased from 13:00 to 19:30 but kept decreasing after 22:00. As the air temperature kept decreasing throughout the sampling period, the water was heated before 24:00 and started to lose heat to the atmosphere afterwards. The alkalinity dropped from 15:00 to 19:30 and increased since 20:00. With a mean value of 2904 µg/L, alkalinity reflected a similar variation trend as $pCO_2$. Like the pH, the conductivity also varied little with the value ranging from 527.7 µS/cm to 540.8 µS/cm. The wind speed was higher in the daytime; the maximum (3.5m/s) was recorded at 16:30, while in the nighttime the sampling point was dominated by calm wind conditions.

**4 Discussion**

**4.1 Damming effect on carbon effluxes in the Upper Mekong River**

[revised manuscript text omitted]

**4.2 Extrapolation of the results and implication for future studies**

The efflux from the pelagic zone and from the littoral zone was 352 mg m$^{-2}$ d$^{-1}$ and 684 mg m$^{-2}$ d$^{-1}$ respectively. Assuming the water level fluctuated frequently within 2.5m and the slope at the bank was 45°, the littoral area covered an area of $1.81 \times 10^5 m^2$. Hence the littoral zone could contribute 6.16t of carbon to the atmosphere, assuming it would be flooded in half of the year. We estimate that the permanent flooded area will be 5,643,000m$^2$ for the GGQ. The carbon dioxide evading from this area will be 200t 
[revised manuscript text omitted]
 Med (Min-Max) | pH Med (Min-Max) | Cond/μS/cm Med (Min-Max) | DO/mg/L Med (Min-Max) | Talk/μg/L Med (Min-Max) | TN/mg/L Med (Min-Max) | TP/mg/L Med (Min-Max) | Chl a/mg/L Med (Min-Max) | $pCO_2$/ppm Med (Min-Max) |
|---|---|---|---|---|---|---|---|---|---|
| R1 | 16.9(8.4-20.5) | 8.40(7.47-8.61) | 355.4(296.2-536.4) | 8.93 (8.08-19.33) | 2608(1696-3036) | 0.51 (0.04-1.40) | 0.12 (0.01-0.73) | 0.99 (0.73-2.34) | 572 (293-4902) |
| R2 | 19.2(8.3-21.1) | 8.35(8.09-8.80) | 295.0(159.8-437.7) | 7.97 (4.61-20.16) | 2508(1888-3456) | 0.69 (0.20-4.47) | 0.30 (0.01-1.65) | 1.15 (0.75-2.09) | 748 (289-1369) |
| P1 | 17.1(8.3-20.5) | 8.38(7.63-8.86) | 352.5(256.6-540.4) | 8.81 (8.03-10.05) | 2486(1712-2928) | 0.51 (0.04-1.66) | 0.04 (0.01-0.65) | 1.01 (0.61-2.68) | 621 (237-3427) |
| P2 | 17.8(8.4-25.0) | 8.35(8.03-8.84) | 330.5(214.2-537.2) | 8.66 (7.94-9.32) | 2338(1528-2928) | 0.59 (0.04-2.30) | 0.02 (0.01-0.52) | 0.92 (0.75-1.68) | 637 (201-1062) |
| P3 | 18.6(8.4-25.0) | 8.28(8.05-8.49) | 333.0(253.2-462.9) | 8.30(7.49-8.83) | 2262(1800-2772) | 0.65 (0.04-1.59) | 0.02 (0.01-0.49) | 0.95 (0.62-1.84) | 698 (448-1257) |
| P4 | 19.6(8.2-25.0) | 8.34(8.08-8.77) | 343.6 (259.4-494.2) | 7.90(7.63-9.87) | 2220(1888-2928) | 0.79 (0.04-2.78) | 0.02 (0.01-0.12) | 0.99 (0.61-1.18) | 747 (188-1183) |
| D | 17.5(8.3-25.0) | 8.37(8.17-8.62) | 340.1 (266.0-529.2) | 9.90(7.96-20.11) | 2508(1784-3000) | 0.52 (0.03-1.88) | 0.02 (0.01-0.71) | 0.99 (0.63-2.05) | 615 (377-958) |
| L | 18.1(8.5-22.1) | 8.34(7.00-8.53) | 357.7(275.4-539.4) | 8.49(6.77-9.07) | 2736(1928-4320) | 0.61 (0.04-2.48) | 0.02 (0.01-0.50) | 0.98 (0.63-1.60) | 750 (353-14764) |

---

## Author Response (AR3)

Author's Response

We really appreciate all the comments and advices from the editor. Thank you very much for your patience in the review and revision of the manuscript. We need to apology for our proofreading on statistics and language. We found some mistakes of the values shown in the abstract, but in the latest version we corrected them and it will not affect the conclusion.

The specific comments are listed below:

Units - starting in the abstract - are the flux units of Carbon or CO2 - consider a unit like mg-C m-2 yr-1, or use a molar unit but be consistent throughout.

Response: All the values of effluxes and graphs in the manuscript were expressed in mg $CO_2$ m$^{-2}$ d$^{-1}$ as illustrated from Fig. 4 to Fig. 10. But we used the flux unit of tons C when calculating the annual emission rates. To make the unit consist, we decide to change the unit of annual emission rates and put the unit as "mg CO2 m-2 d-1" and "mg CO2 yr-1" to avoid confusion.

Figure 7 . change incontinuous to discontinuous to match figure legend.

Response: We changed it as suggested.

Pg 3 line 19 - how many dams in china? can you state?

Response: We added the number into the article.

Pg 1 Line 18 - check grammar - pelagic area...resulted from (please check grammar throughout)

Response: Very sorry for the grammar. We have corrected some mistakes and tried our best to examine the grammar.